# Benchmarking the translational potential of spatial gene expression prediction from histology

Chuhan Wang [1,2,3,7], Adam S. Chan [1,4,5,7], Xiaohang Fu [1,2,3,4,5], Shila Ghazanfar [1,4,5], Jinman Kim[1,2,3], Ellis Patrick [1,3,4,5,6,8] ✉ & Jean Y. H. Yang [1,3,4,5,8] ✉

Spatial transcriptomics has enabled the quantification of gene expression at spatial coordinates across a tissue, offering crucial insights into molecular underpinnings of diseases. In light of this, several methods predicting spatial gene expression from paired histology images have provided the opportunity to enhance the utility of obtainable and cost-effective haematoxylin-and-eosin-stained histology images. To this end, we conduct a comprehensive benchmarking study encompassing eleven methods for predicting spatial gene expression with histology images. These methods are reproduced and evaluated using five Spatially Resolved Transcriptomics datasets, followed by external validation using The Cancer Genome Atlas data. Our evaluation incorporates diverse metrics which capture the performance of predicted gene expression, model generalisability, translational potential, usability and computational efficiency of each method. Our findings demonstrate the capacity of the methods to predict spatial gene expression from histology and highlight areas that can be addressed to support the advancement of this emerging field.

Spatially resolved transcriptomics (SRT) data captures the spatial organisation and heterogeneity of genes in tissues at high resolution, fundamentally transforming the way we study and understand biological processes. It has enabled the identification of key genes and pathways involved in intricate mechanisms underlying diverse biological processes and complex cellular phenomena[1–4]. While these new SRT technologies hold promise for producing new insights, they remain costly, preventing their direct adoption into routine clinical use at this stage. In contrast, haematoxylin-and-eosin-stained (H&E) histopathology images are more cost-effective and are routinely used in clinical practice. Thus, if we can leverage modern machine learning together with information from SRT technologies to predict in silico spatial profiles of gene expression values from paired histology images of the same tissue, we could enhance information gained from existing H&E images. This would facilitate large-scale examinations of spatial gene expression variations and lead to the potential discovery of biomarkers and therapeutic targets for complex diseases.

To date, several approaches have been developed to predict in silico spatially resolved gene expression (SGE) patterns using H&E data alone (Fig. 1a, Supplementary Table 1). Among these methods, Convolutional Neural Networks (CNN) and Transformers are commonly selected architectures for extracting local and global 2D vision features around each sequenced spot from corresponding histology image patches. Some methods further implemented extra components, including

[1]Sydney Precision Data Science Centre, University of Sydney, Sydney, NSW, Australia. [2]School of Computer Science, The University of Sydney, Sydney, NSW, Australia. [3]Laboratory of Data Discovery for Health Limited (D24H), Hong Kong SAR, China. [4]School of Mathematics and Statistics, The University of Sydney, Sydney, NSW, Australia. [5]Charles Perkins Centre, The University of Sydney, Sydney, NSW, Australia. [6]The Westmead Institute for Medical Research, Sydney, NSW, Australia. [7]These authors contributed equally: Chuhan Wang, Adam S. Chan. [8]These authors jointly supervised this work: Ellis Patrick, Jean Y. H. Yang. ✉e-mail: ellis.patrick@sydney.edu.au; jean.yang@sydney.edu.au

**a** **Predicting gene expression from histology using machine learning**

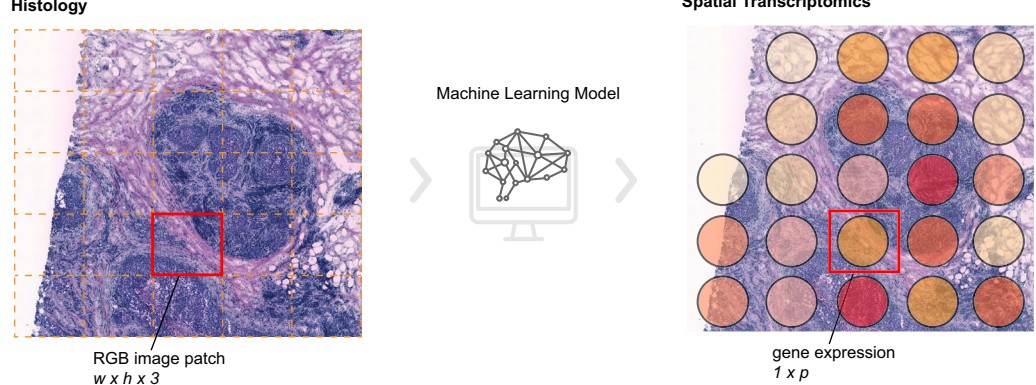

**b** **Benchmarking methods predicting ST from Histology**

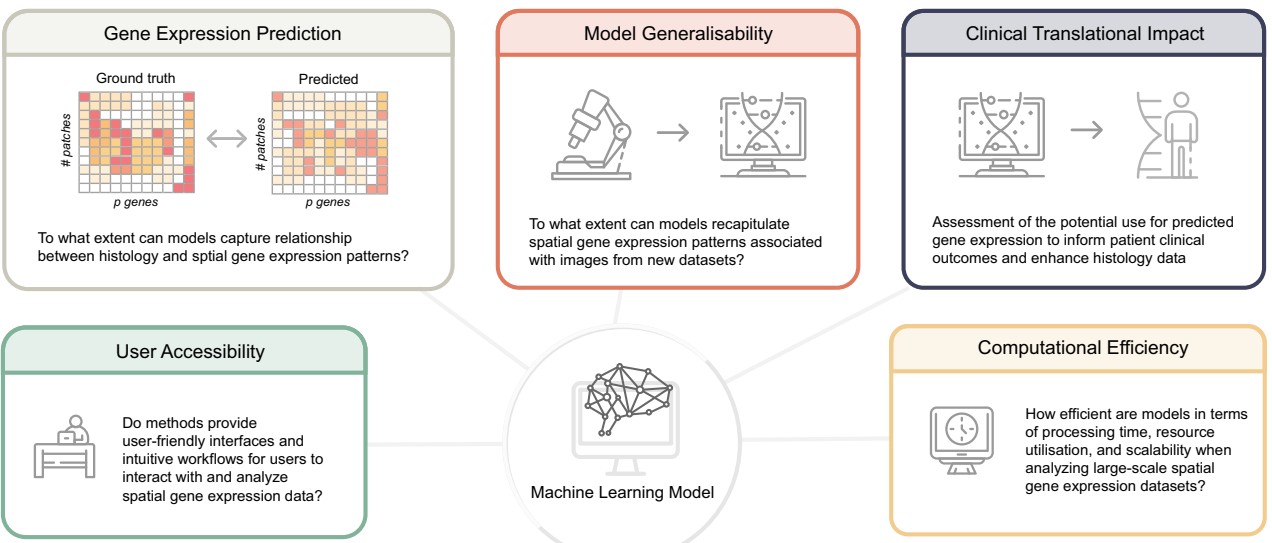

**Fig. 1 | Overview of the benchmarking process and the key aspects of the evaluation. a** Illustration of the machine learning task of predicting spatial gene expression from H&E images. **b** Overview of the evaluation categories used in the benchmarking of each method. Figure generated using Adobe Illustrator, with some graphics adapted from Adobe Stock under an Education License.

super-resolution enhancement of SGE and Graph Neural Networks (GNN) to capture neighbourhood relationships between adjacent spots. Furthermore, an exemplar module had been used to guide predictions by inferring from the gene expression (GE) of the most similar exemplars. These learned features are then used to predict SGE that enables image-based screening for molecular biomarkers with spatial variation. As these methods were recently published in the last 3 years and are continuously evolving, their performance has not yet been comprehensively benchmarked. While there exist general reviews on various aspects of SRT analysis[5–7], they have only surveyed and categorised methods predicting SGE from H&E without any benchmarking.

Evaluation studies that are presented when proposing a new method are limited in nature and often not independent. Among the current literature, methods for predicting SGE from H&E have been mainly assessed and evaluated for their efficacy by their developers. Evidently, different evaluation frameworks have been used to assess the methods. For instance, ST-Net[8], DeepSpaCE[9], DeepPT[10] and TCGN[11] evaluated the performance of their models on external The Cancer Genome Atlas (TCGA)[12] data to determine the generalisability of their models. Hist2ST[13], EGNv1[14], EGNv2[15], TCGN and THItoGene[16] performed an ablation study to assess the contributions of different components

of their deep-learning architecture. BrST-Net[17] conducted a comparison of 10 state-of-the-art CNN and transformer architectures to determine the effectiveness of the backbone models. iStar[18] demonstrated the application of its predictions by performing tissue region segmentation. Overall, there is no consistent and detailed evaluation framework comparing the performance of these approaches in predicting in silico SGE, due in part to the recent development of this field, limited data availability and the complexity of executing these methods. Consequently, we lack a comprehensive understanding of the performance, stability and the broader applicability of individual methods. Additionally, there is a lack of critical evaluation regarding how different spatial characteristics of H&E images can affect the accuracy of SGE predictions, especially on their impact on downstream analysis as well as potential clinical translation.

To this end, we provide a benchmark and review of methods predicting SGE from H&E images. We compared the performance of eleven methods using 28 different metrics broadly grouped into five categories. These include metrics that focus on the evaluation of spatial and biological characteristics of genes that are predicted effectively. Metrics also include the impact of prediction performance on downstream applications. Given the ongoing development in

**Table 1 | Architecture characteristics of methods benchmarked**

| Model | Platform | Deep-learning model | Architecture characteristics | | | |
|---|---|---|---|---|---|---|
| | | | Local features (one spot)[a] | Local + global features (spot-neighbourhood relations)[b] | Global features (spot-spatial relations)[c] | Trainable parameters |
| ST-Net | Python | Yes | Pretrained DenseNet 121 | NA | NA | 8 M |
| HisToGene | Python | Yes | Learnable linear layer | Super Resolution | ViT | 222 M |
| DeepPT | Python | Yes | Pretrained ResNet50 + Autoencoder + MLP | NA | NA | 26 M |
| Hist2ST | Python | Yes | Convmixer | GNN - GraphSAGE | Transformer | 230 M |
| DeepSpaCE | R & Python | Yes | VGG16 | Super Resolution | NA | 137 M |
| GeneCodeR | R | No | NA | NA | NA | 75 M |
| EGNv1 | Python | Yes | ViT + Exemplar Extractor (ResNet) | NA | Exemplar | 135 M |
| EGNv2 | Python | Yes | Exemplar Extractor (ResNet) + GCN | NA | Exemplar + Graph Construction | 12 M |
| TCGN | Python | Yes | CNN + ViT + GNN | NA | NA | 29 M |
| THItoGene | Python | Yes | Dynamic Convolution + Efficient-Capsule Module | Graph Attention Network (GAT) | ViT | 63.6 M |
| iStar | Python | Yes | HViT (super-resolved features at near-single-cell level) | NA | HViT | 2.19 M |

[a]Local feature within a patch.
[b]Neighbouring global/contextual features within a section of histology image.
[c]Global/contextual features within the whole histology image.

predicting SGE from H&E providing exciting potential for clinical usage, our benchmark study provides an analysis of significant achievements of existing methods, demonstrating their practical impact. Importantly, we provide a vital assessment of the limitations and generalisability of their translational potential. Ultimately, we offer guidance for new users of these methods and highlight current challenges for developers aiming to advance methodologies. Furthermore, our study can serve as a valuable tool, facilitating the rapid and independent evaluation of future methods.

## Results
### Benchmarking in silico spatial gene expression prediction from H&E image
To assess current methods of predicting gene expression from histology images, we have developed a benchmarking framework across five key categories (Fig. 1b). We employed a hierarchy of evaluation categories: (1) within image SGE prediction performance for lower-resolution spatial transcriptomics (ST) data initially described by Stahl et al. and higher-resolution 10x Visium data; (2) cross-study model generalisability, evaluated by applying models trained on ST data to predict Visium tissues, as well as to predict TCGA images to identify whether models were useful for predicting existing H&E images; and (3) clinical translational impact by predicting survival outcomes and canonical pathological regions via predicted SGE. In addition, we considered (4) the usability of the methods encompassing code, documentation and the manuscript; (5) and the computational efficiency. These categories provide the foundation for comprehending the accuracy and applicability of histology-based gene expression prediction methods. We employed an evaluation strategy that utilised 28 metrics across all evaluation categories to reveal diverse characteristics of eleven methods predicting SGE from histology images (Table 1). Our findings highlighted that no single method emerged as the definitive top performer across all categories of evaluation. HisToGene[19], DeepSpaCE and Hist2ST demonstrated notable performance in model generalisability and usability (Fig. 2a), while EGNv2 and DeepPT exhibited the highest accuracy in predicting SGE for ST and Visium data respectively, they showed limitations in distinguishing survival risk groups and in model generalisability based on the predicted SGE. In the following section, we will examine the various categories (Fig. 2b) in detail.

### Most methods can capture biologically relevant gene patterns from tissue images using ST datasets
Several evaluation metrics were used to identify methods with superior SGE predictive performance. After models were trained consistently to predict SGE from histology, we compared the predicted SGE to the ground truth SGE in hold-out test images from cross-validation (CV) (Fig. 2c). Across the HER2+ and cutaneous squamous cell carcinoma (cSCC) ST datasets, we measured the relative performance across all genes based on Pearson Correlation Coefficient (PCC), Mutual Information (MI), Structural Similarity Index (SSIM) and Area Under the Curve (AUC) (Table 2) for each method (Fig. 2d). Notably, EGNv2 had the best overall performance (PCC = 0.28; MI of 0.06; SSIM of 0.22; AUC of 0.65), suggesting that its predicted SGE and spatial patterns were more aligned to the ground truth SGE patterns than other methods, which potentially benefits from the SGE inference from the most similar spots. Hist2ST had the second highest average MI of 0.06 and AUC of 0.63, suggesting the predicted SGE displayed some alignment with the ground truth and was able to distinguish between zero and non-zero expressions better than other methods. Since we observed PCC and SSIM are low crossing all methods, we tried to determine how well each method extracted biologically relevant insights from H&E images in HER2+ ST. We assessed this by looking into selected highly variable genes (HVG) and spatially variable genes (SVG) (Fig. 3a) and compared the performance across all genes (Supplementary Figs. 1 and 2) using one-sided Wilcoxon rank-sum test. For both HVGs and SVGs, most methods exhibit higher correlation (DeepPT, GeneCodeR[20], EGNv1, EGNv2 and iStar $p < 0.05$ under HVGs and all methods except Hist2st, $p < 0.05$ under SVGs) or SSIM (all methods except HisToGene and TCGN, $p < 0.05$ under HVGs and all methods, $p < 0.05$ under SVGs) compared to using all genes, and we believe this provides a more meaningful evaluation of methods. Interestingly, DeepPT, characterised by a simpler CNN architecture, demonstrated predictive advantages compared to Hist2ST, which additionally captures the global spatial features on the whole image slide using the learned features from each spot.

Subsequently, we examined if the genes being accurately predicted by the methods were specific to the tissue types in the data. Here, we assessed the capacity for methods to capture tissue-relevant SGE by examining the biological relevance of the top correlated genes (Supplementary Tables 3 and 4). Here, EGNv2 observed the highest

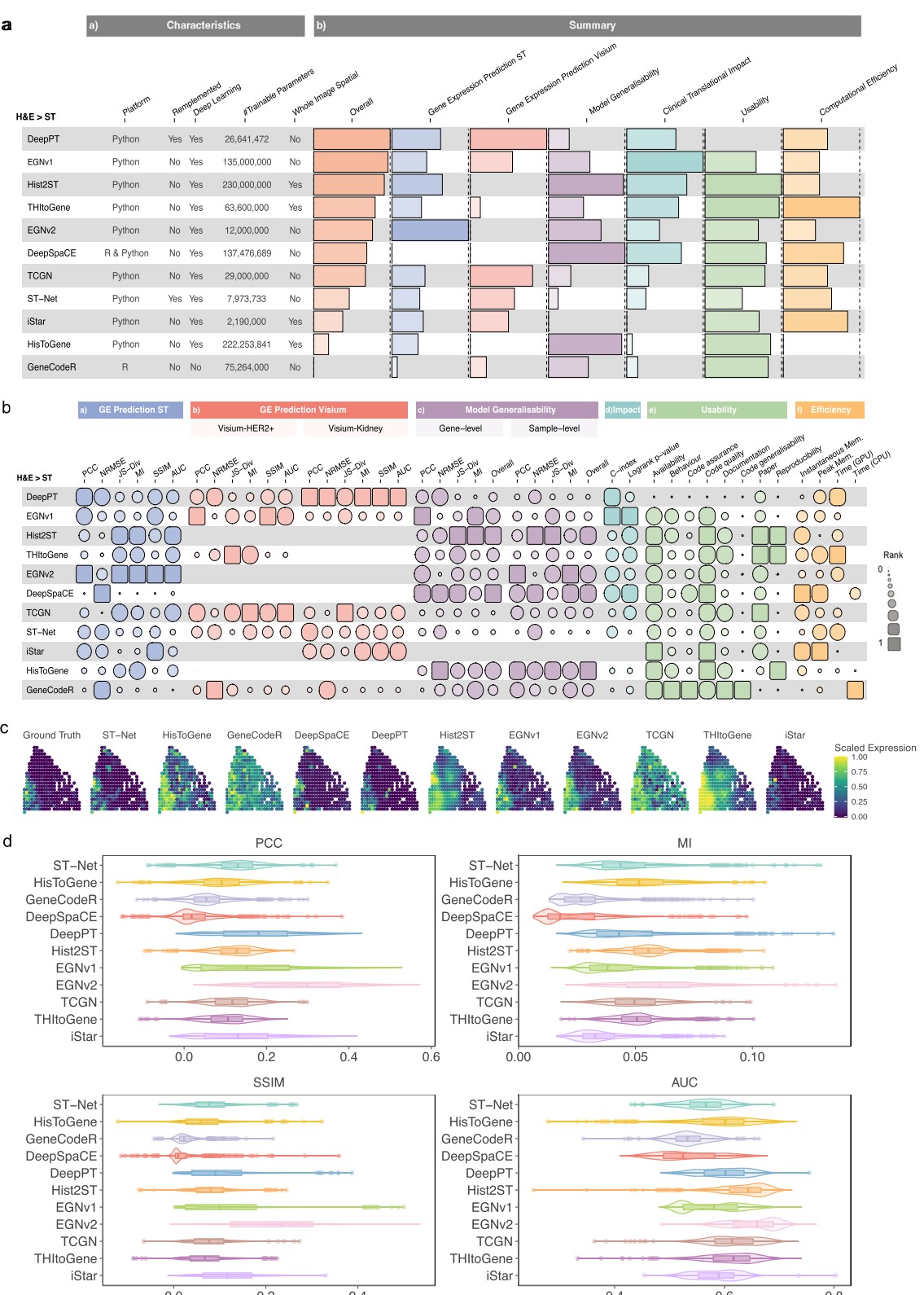

**Fig. 2 | Summary of various characteristics of the selected methods and their overall evaluation results. a** Summary heatmap of methods predicting spatial gene expression from H&E images highlighting key characteristics and ranking their performances under each evaluation category. **b** Detailed heatmap of rankings of each method under each evaluation metric grouped by category. **c** Spatial plots of FASN expression for one sample B1 in the HER2+ ST dataset. **d** Boxplot/violin plots of the average PCC, MI, SSIM and AUC between the ground truth gene

expression and predicted gene expression. Metrics measured from the test fold of a 4-fold CV, averaged over each gene across HER2+ ($n = 785$) and cSCC ($n = 997$) ST datasets. The bounds of the box correspond to the 25th percentile (first quartile) and 75th percentile (third quartile). The line within the box represents the median. The boxplot's lower whisker extends 1.5 times the interquartile range below the first quartile, while the upper whisker extends 1.5 times the interquartile range above the third quartile. Source data are provided as a Source Data file.

**Table 2 | Evaluation metrics used and their interpretations**

| Category | Metric name | Full metric name | Explanation |
|---|---|---|---|
| Gene expression prediction/ Model generalisability | PCC | Pearson Correlation Coefficient | Measures the linear relationship between predicted and observed gene expression, providing a value between −1 and 1, where 1 indicates a perfect positive correlation and −1 indicates a perfect negative correlation. |
| | MI | Mutual Information | Measures the amount of information shared between predicted and observed gene expression, capturing their statistical dependence. Higher values indicate stronger dependence and similarity between the variables. |
| | JS-Div | Jensen–Shannon divergence | Quantifies the dissimilarity or divergence between the predicted and true gene expression probability distributions. It provides a measure of dissimilarity that ranges from 0 to 1, where 0 indicates identical distributions and 1 indicates complete dissimilarity. A lower JS-Div indicates better agreement and similarity between the distributions. |
| | NRMSE | Normalised Root Mean Squared Error | The RMSE (Root Mean Squared Error) between predicted and observed gene expression values, normalised by the range of the observed values. It provides a normalised measure of prediction accuracy, allowing for comparison across different datasets or scales. A lower NRMSE indicates better prediction performance. |
| | SSIM | Structural Similarity Index | Evaluates the structural similarity of spatial patterns between predicted and observed gene expression by treating each spot as a 'pixel' in the spatial grid. It measures the similarity of intensities, luminance, contrast and structural information. Higher SSIM values indicate better structural similarity. |
| | AUC | Area Under the Curve | Quantifies how well the predicted gene expression can discriminate between binarisation of zero vs. non-zero (or small vs. large) values of the observed gene expression values. It ranges from 0 to 1, and an AUC of 1 suggests that the predicted gene expression values can perfectly discriminate between the binarisation of observed gene expression value. |
| Clinical translational impact | C-index | Concordance index | Quantifies the discriminatory power of a predictive survival model by assessing its ability to correctly rank or classify pairs of observations, typically in terms of their survival times or outcome probabilities. A C-index value of 0.5 indicates random chance, while a value of 1.0 signifies perfect discrimination. |
| | Log-rank $p$ value | Log-rank test $p$ value | The log-rank $p$ value is a statistical measure commonly used in survival analysis to assess the difference in survival or event occurrence between two or more groups. If the $p$ value is small (typically below a predefined significance level, such as 0.05), it suggests that the observed differences in survival curves are unlikely to have occurred due to chance alone. |

correlations in genes *GNAS* ($r = 0.47$) and *FASN* ($r = 0.46$) in HER2+ ST dataset (Fig. 3b) and *PFN1* ($r = 0.53$) in cSCC ST dataset (Fig. 3c). In the context of HER2+ breast cancer, *FASN* is known as a hallmark feature observed early in most human carcinomas and precursor lesions which has been associated with adverse patient outcomes and therapeutic resistance[21]. This gene is one of the highest predicted genes ($r = 0.30$ in HER2+ ST) identified across all methods and is aligned with key cancer-related molecular pathways. Additionally, *MYL12B*, with an average correlation of 0.24, is known to be involved in the regulation of cell morphology with links to cancer progression[22]. In the cSCC ST dataset, *LMNA* ($r = 0.22$), has been shown to have increased expression in skin cancer[23]. These findings indicate that all methods, despite the relatively low average correlation, were able to capture biologically relevant gene patterns from tissue images.

We used the predicted SGEs to perform K-means clustering and identify spatial regions in eight H&E images from HER2+ ST samples (Supplementary Fig. 3). Interestingly, the clustering results based on Ground Truth SGE did not always outperform those generated using predicted SGE. This suggests that the predicted SGE from H&E images captures additional imaging features from each spot and its surrounding tissue, providing a more comprehensive view of spatial patterns that enhance tissue region identification.

Furthermore, we delved into the capacity of the methods to discriminate between different levels of gene expression, employing several binary thresholds (Fig. 3d). Our findings revealed that the predicted GE across all methods was able to distinguish between ground truth GE counts greater than 10 and less than 10 with a mean AUC = 0.65. This outperformed the ability to predict between non-zero and zero GE where mean AUC = 0.57, suggesting that methods were more capable of distinguishing between higher and lower values of GE. We also found that all methods had a negative correlation between the predicted vs ground truth correlation and the percentage of zeros for each gene ($p < 2 \times 10^{-5}$), which was more notable for DeepPT, GeneCodeR, ST-Net, DeepSpaCE, EGNv1, EGNv2 and iStar (Supplementary Fig. 4). For the top performing methods in GE prediction, EGNv2 and DeepPT, the performance was noticeably affected by the variance, absolute expression and percentage of zeros of the genes it was trained on. Thus, we observed that some methods perform better when noisy genes are removed from the analyses.

**Characteristics of methods play a large role in gene-specific performance**

We set out to uncover the role of varying deep-learning architectures in gene-specific prediction performance and the extent to which these structures shaped our results. In Fig. 3e, we plotted the average correlation of each gene over each method in the HER2+ ST dataset. We then employed hierarchical clustering on the correlations from each gene and method to identify the similarities between method performance based on characteristics of their deep-learning architectures. Notably, we observed that EGNv2, ST-Net and DeepPT clustered together, a finding consistent with their shared utilisation of pretrained CNNs focusing on single image patches feature only (Table 1). Hist2ST and HisToGene formed another distinct cluster, then further clustered with TCGN and THItoGene. TCGN applied transformer and GNN on a single spot to optimise spot-level global features, while the rest of the methods attributed to their emphasis on global interactions and their models taking whole slide image features as input as opposed to just image patches in the other methods. Based on the predicted SGE performance in the ST datasets, we found that the methods which were members of

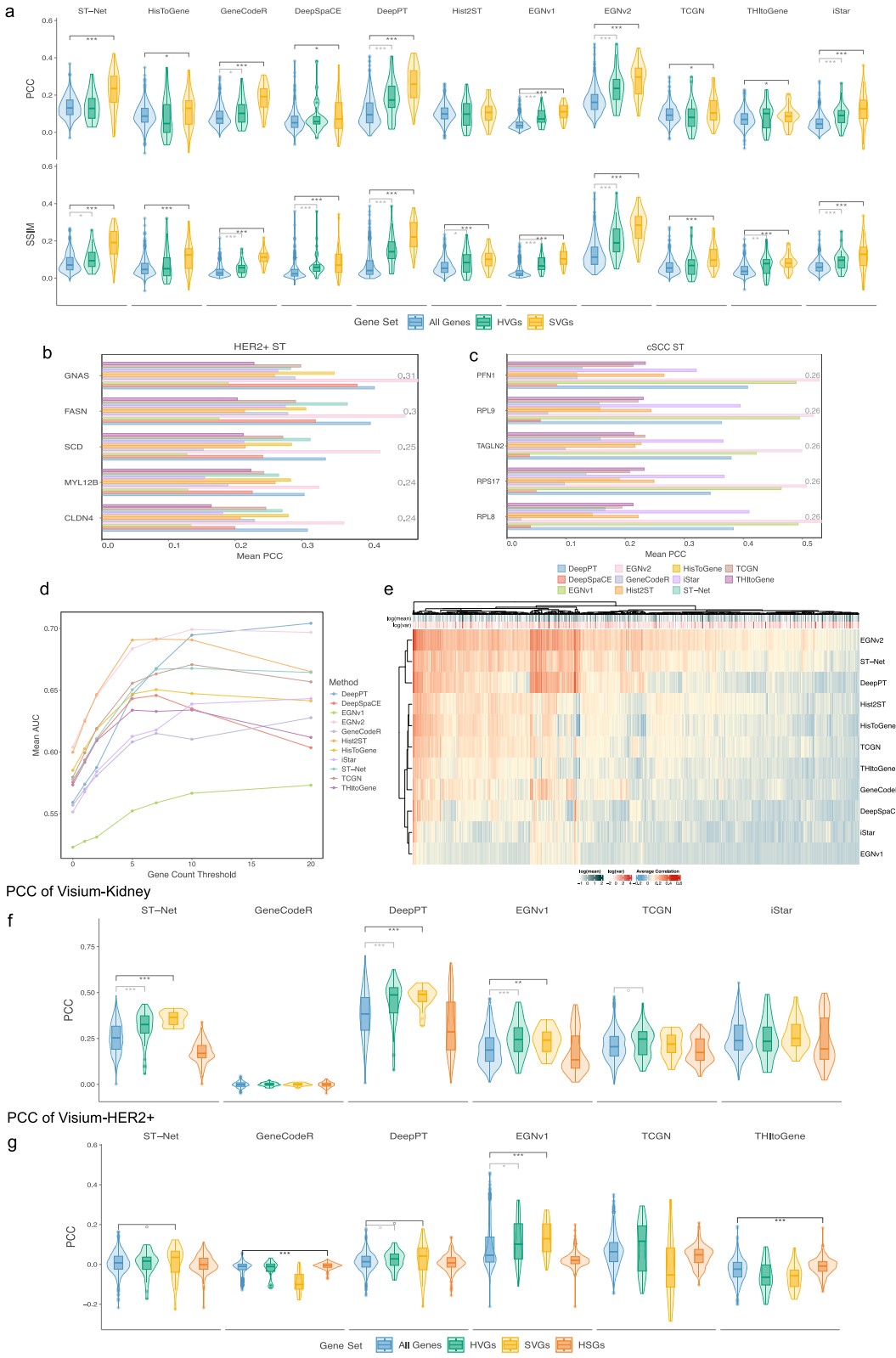

PCC of Visium-Kidney

PCC of Visium-HER2+

the cluster comprising EGNv2, ST-Net and DeepPT outperformed the others. This suggests that methods focusing on extracting image features within patches using CNN-based architectures are particularly well-suited for this task. The consistent distribution of prediction performance across genes highlights the influence of the structural design of these methods. This finding provides guidance for architecture selection in future method development.

Furthermore, we aimed to determine differences in gene-specific performance due to different data processing between methods. A substantial proportion of genes displayed positive correlations across all tested methods (Fig. 3e), highlighting common genes that were more generally well-predicted such as *MYL12B*, *SCD* and *GNAS*. We also identified a subset of genes that exhibited strong performance in DeepPT and ST-Net but displayed diminished performance in other

**Fig. 3 | Investigation of correlations and performance variation among methods in the ST and Visium datasets. a** PCC and SSIM violin and boxplots for each method in HER2+ ST dataset for all genes (n = 785), HVGs (n = 30) and SVGs (n = 20 per image sample). Significance between HVGs, SVGs and all genes, is calculated using one-sided Wilcoxon rank-sum test (The significance levels were defined as °p < 0.1, *p < 0.05, **p < 0.01, ***p < 0.001). Exact p values are included in the Source Data file. The bounds of the box correspond to the 25th percentile (first quartile) and 75th percentile (third quartile). The line within the box represents the median. The boxplot's lower whisker extends 1.5 times the interquartile range below the first quartile, while the upper whisker extends 1.5 times the interquartile range above the third quartile. **b** Top five correlated genes in HER2+ ST and **c** cSCC ST dataset, with grey numbers indicating average test set correlations. **d** AUC plot of predicted gene expression distinguishing ground truth binarized at various thresholds.

(x-axis). **e** Heatmap of average correlation of each gene in HER2+ ST dataset and each method. The log of the mean and variance of each gene are coloured above the heatmap. PCC violin and boxplots for each method in **f** Visium-Kidney data and **g** Visium-HER2+ data for all genes (n = 992, 990), HVGs (n = 48, 36), SVGs (n = 20, 20 per image sample) and HSGs (n = 145, 274). Significance between HVGs, SVGs, HSGs and all genes is calculated using one-sided Wilcoxon rank-sum test (The significance levels were defined as °p < 0.1, *p < 0.05, **p < 0.01, ***p < 0.001). Exact p values are included in the Source Data file. The bounds of the box correspond to the 25th percentile (first quartile) and 75th percentile (third quartile). The line within the box represents the median. The boxplot's lower whisker extends 1.5 times the interquartile range below the first quartile, while the upper whisker extends 1.5 times the interquartile range above the third quartile. Source data are provided as a Source Data file.

methods such as *C3* and *BGN*. This occurred due to an observed reduction in correlation between the original GE data and the normalised data used for model training, from normalisation techniques specific to the Hist2ST and HisToGene methods (Supplementary Fig. 5). These correlation findings highlight an interplay between methods of input processing and SGE gene-specific prediction performance.

## Spatial resolutions and gene matrix sparsity influence methods' performance

We expanded our evaluation to examine data with higher spatial resolutions by adding three additional Visium datasets. When evaluating based on the Visium-Kidney dataset (Supplementary Fig. 6), a non-tumour tissue, Fig. 3f shows that DeepPT achieved the highest average correlation of HVG of 0.45 and SVG of 0.47, followed by ST-Net (HVG = 0.31, SVG = 0.36) and EGNv1 (HVG = 0.25, SVG = 0.24). The strong performance of DeepPT and ST-Net (CNN-based methods) on non-tumour tissue is likely due to the low tissue heterogeneity of the sample, where patches containing a spot provided sufficient information for learning, making additional neighbourhood information less valuable due to the uniformity of surrounding tissues. In general, we found that a number of methods struggled with Visum data. These methods showed almost no variation in predicted gene expression across spots when training using Visium-Kidney and Visium-Hercep-Test2+, (Supplementary Figs. 7 and 8) data, which did not happen with ST datasets. One potential reason is due to the difference in gene matrix sparsity, where Visium data typically has more densely expressed genes compared to ST datasets (Supplementary Fig. 9). We are able to resolve this issue for two methods (His2ST; THItoGene) by selecting high-sparsity genes (HSGs) for training (Supplementary Figs. 10 and 11) but found that it did not improve overall prediction performance.

Next, we examined the flexibility of these methods by performing cross-study evaluation. Firstly, we assessed performance across tumour subtypes, where we trained on Visium-Hercep-Test2+ and tested on Visium-HER2+ (Supplementary Fig. 12). EGNv1 demonstrated significant improvement in average correlation using HVGs and SVGs of 0.11 and 0.13, respectively (Fig. 3g), their performance is in general weaker than the within study evaluation. This suggests that models trained on a specific breast cancer subtype struggle to generalise to others, as the biological characteristics differ between subtypes. Secondly, we evaluated prediction transferability across different spatial resolutions using models (best fold) trained on HER2+ ST data, directly testing them on a Visium-HER2+ breast cancer patient (Supplementary Fig. 13). Most methods showed limited effectiveness, except for HisToGene. HisToGene, originally trained using super-resolution with smaller tissue patches, allows the model to capture features in the scale that is more similar to Visium data. These results highlight the need for improved adaptation strategies to accommodate different platforms with varying imaging resolutions.

## Comparing the translational potential across methods using TCGA-BRCA data

We investigated the translational potential of predicted SGE for diagnostics in pathology as one of the downstream applications of predicting SGE from H&E. We evaluated this from three perspectives: (1) robustness to input quality; (2) generalisability to an independent cohort; and (3) utility in survival prognosis. Firstly, we assessed whether H&E image quality impacted the accuracy of prediction. For the TCGA-BRCA stage I breast cancer (BC) H&E images, we first calculated a wide range of H&E quality control (QC) metrics (Supplementary Fig. 14) using HistoQC[24]. Next, we applied the best fold models, selected based on validation set performance and pretrained on the HER2+ ST dataset, to predict bulk RNA-Seq gene expression patterns for these images. We used the patient-level correlation as a measure of prediction performance by calculating the correlation between predicted GE pseudobulk and bulk GE for each image. We then measured the correlation between the first principal component of the H&E QC metrics and prediction performance over each image and method. Previous literature[25] had shown that the quality of H&E impacts segmentation performance and so we expect that a robust method will exhibit minimal dependence between image quality and the SGE prediction performance. DeepPT showed an undesirably high correlation (r = 0.53) between image quality and prediction performance (Fig. 4a). Meanwhile, HisToGene (r = 0.05) and Hist2ST (r = 0.19) had little to no dependence on the obtained QC metrics, indicating their robustness to image quality. HisToGene and Hist2ST used normalisation in their workflow, which indicates that input processing may help reduce dependence on image quality. Hence, it is crucial to take image quality into account during method development, as it can have unintended effects on performance.

We further assessed the generalisability of all methods by determining whether methods trained on ST data could feasibly be used on an independent set of H&E images. Using the predicted pseudobulk GE from each method and the bulk GE associated with the TCGA-BRCA H&E images, we calculated the patient-level correlations as defined above. The patient-level correlations averaged across methods were then used to compare the translational performance (Fig. 4b). The methods HisToGene, EGNv2 (mean = 0.53) and DeepSpaCE (mean = 0.52) performed the best in this section. These methods were followed by GeneCodeR, TCGN (mean = 0.49), Hist2ST (mean = 0.44), EGNv1 (mean = 0.42), THItoGene (mean = 0.39), DeepPT (mean = 0.37) and ST-Net (mean = 0.33), suggesting that all methods were able to capture the overall trends and variations in GE for the majority of patients. The positive patient-level correlations suggest that the methods have utility in providing GE information for histology images that do not necessarily have associated SGE data.

We then examined the downstream utility of predicted pseudobulk GE from H&E in breast cancer survival analysis in the TCGA-BRCA images. For each method, we built multivariate Cox regression models from the pseudobulk GE to predict survival for three subtypes of

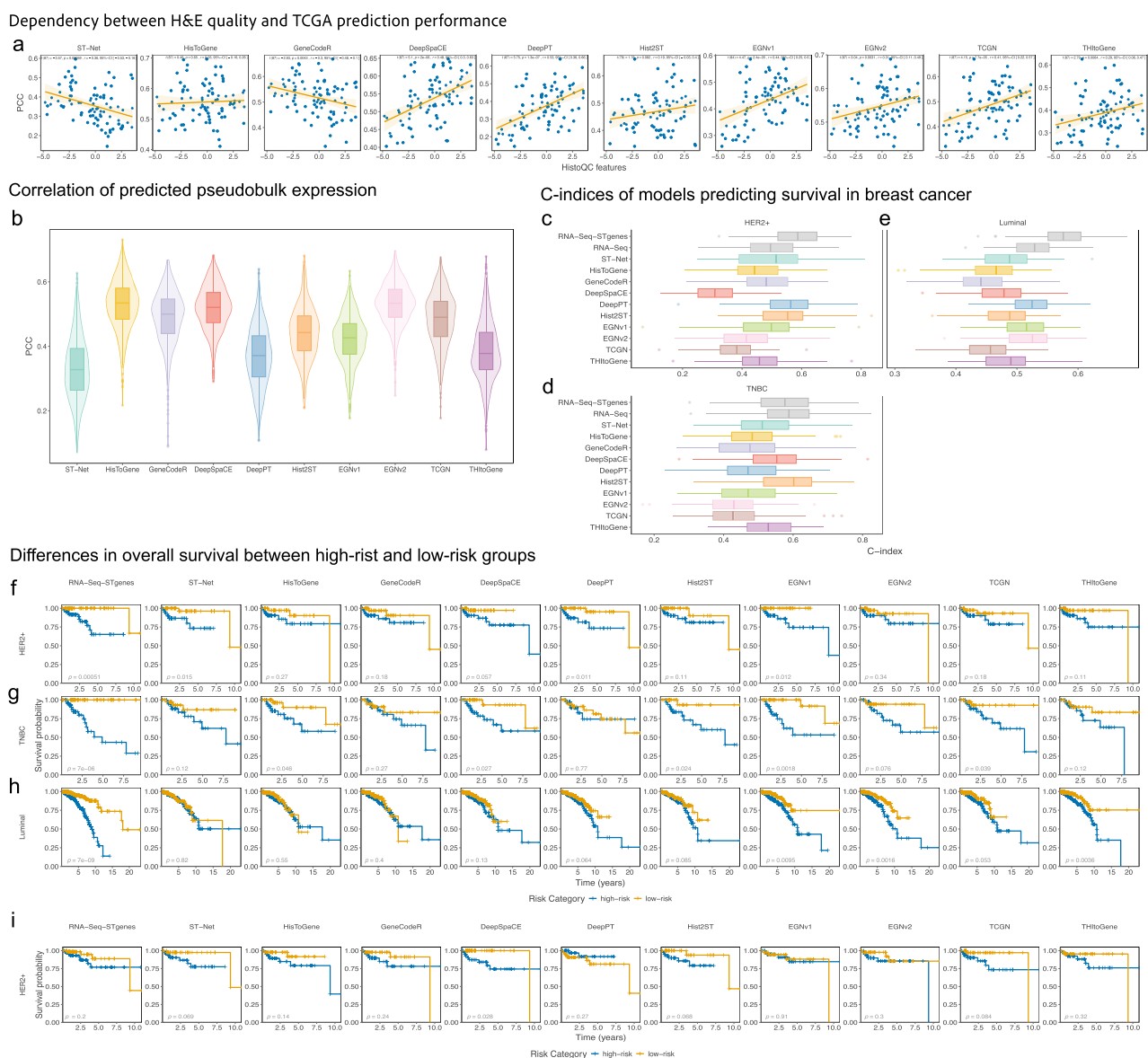

**Fig. 4 | Evaluation of translational potential of gene expression predictions.** **a** Scatterplot of patient-level correlations between predicted pseudobulk gene expression and RNA-Seq bulk gene expression vs. first principal component of calculated H&E quality control metrics. H&E QC metrics were calculated on stage I breast cancer patients. The solid line represents the line of best fit, while the error band represents the 95% confidence interval (CI). Correlation test statistics, degrees of freedom, $p$ values, effect sizes and CIs are shown on each panel. **b** Boxplot of patient-level correlations between predicted pseudobulk gene expression and RNA-Seq bulk gene expression over all analysed TCGA-BRCA images ($n = 671$) split by method. **c–e** C-indices of TCGA-BRCA patients survival prediction, using RNA-Seq bulk, RNA-Seq bulk using only genes present in HER2+ ST dataset and the predicted pseudobulk from each method. C-indices were calculated from the test sets of a 3-fold CV with 100 repeats ($n = 100$) trained within **c** HER2+ ($n = 92$), **d** luminal

($n = 463$) and **e** TNBC ($n = 79$) breast cancer clinical subtypes. The bounds of the box in (**b–e**) correspond to the 25th percentile (first quartile) and 75th percentile (third quartile). The line within the box represents the median. The boxplot's lower whisker extends 1.5 times the interquartile range below the first quartile, while the upper whisker extends 1.5 times the interquartile range above the third quartile. Outliers are shown as individual data points. **f–i** Kaplan–Meier curves for patients split into high and low-risk groups by the median risk predictions for **f** HER2+, **g** luminal and **h** TNBC breast cancer subtypes. Models were trained on all patients and then the predictions of the training patients were used. The $p$ value represents the result of the two-sided log-rank test for assessing the statistical significance of differences in survival between the groups. **i** Kaplan–Meier curves constructed using the median risk prediction from a 3-fold CV with 100 repeats within the HER2+ breast cancer subtype. Source data are provided as a Source Data file.

breast cancer (BC): HER2+ BC, triple-negative breast cancer (TNBC) and luminal BC. We compared these models against models built using the bulk RNA-Seq (baseline) data using average C-index from a 3-fold CV with 100 repeats. In HER2+ BC patients, DeepPT had the highest average C-index = 0.55 compared to 0.58 for the RNA-Seq (Fig. 4c). For TNBC, Hist2ST had the highest average C-index of 0.58, compared to 0.57 for RNA-Seq (Fig. 4d). For luminal BC, DeepPT and EGNv2 had the highest C-index of 0.52 compared to 0.58 for RNA-Seq (Fig. 4e). Across all BC subsets, Hist2ST had the highest average C-index from

evaluating using CV and DeepPT had the highest C-index from evaluating using the training data (Supplementary Fig. 15). Subsequently, we binarised the risk model predictions into high and low-risk groups through using median prediction to split patients then constructed Kaplan–Meier (KM) survival curves. Following the analysis of Xu et al.[26], we were able to capture the same distinct KM curves for the RNA-Seq models in each BC subset of patients ($p \leq 5.1 \times 10^{-4}$) using the risk predictions of the patients the models were trained on (Fig. 4f–h). However, we also showed that under CV (Supplementary Fig. 16) the

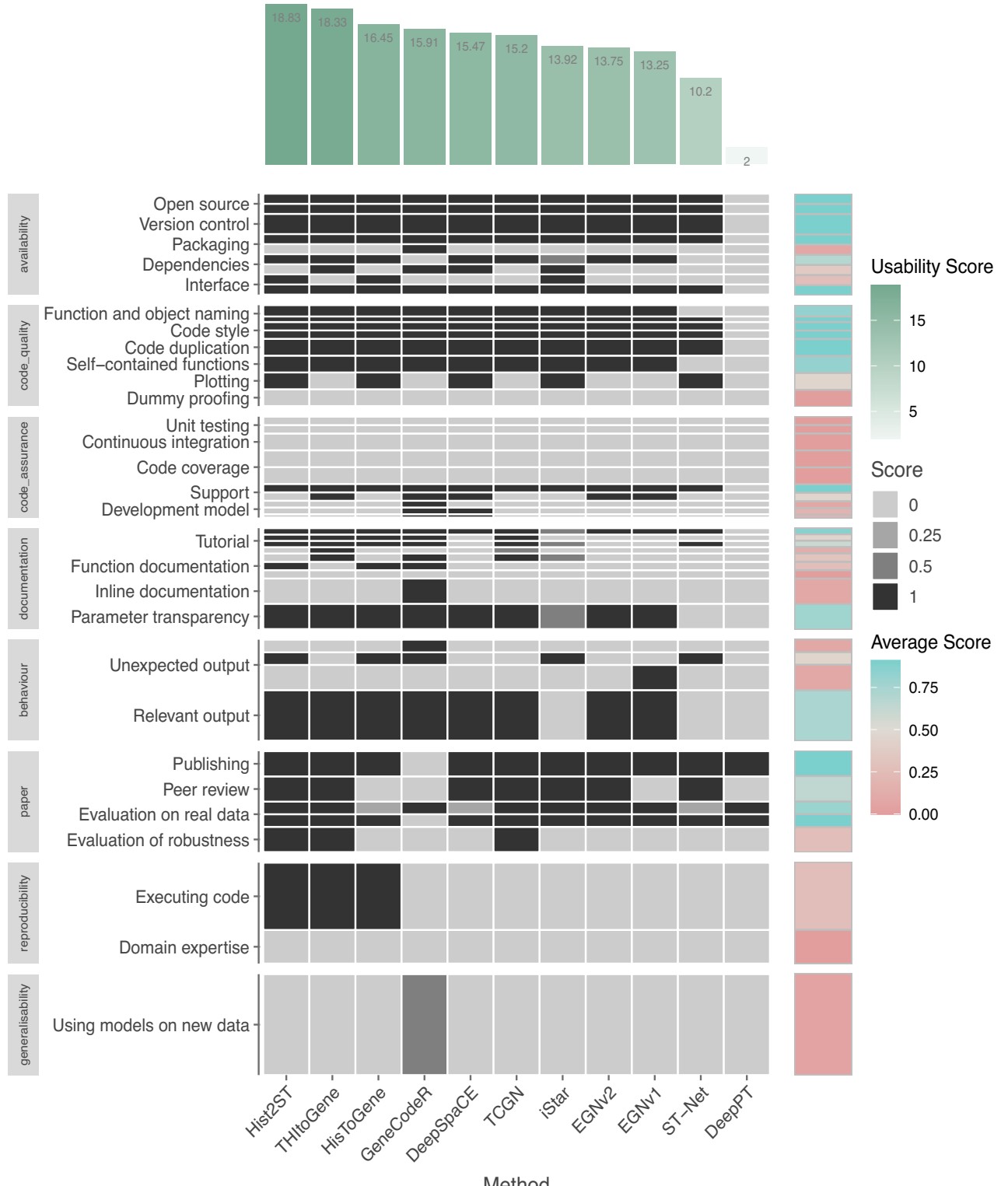

**Fig. 5 | Usability score for the selected methods.** Heatmap showing the scores of each method under each usability scoring category. Height of each box represents the weight of each criterion contributing to the overall score plotted in the green bar plot on top. The blue-pink heatmap represents the row-wise average of the category scores over all methods. Source data are provided as a Source Data file.

distinction between high and low-risk groups was no longer significant except for DeepSpaCE ($p = 2.8 \times 10^{-2}$), Hist2ST ($p = 6.8 \times 10^{-2}$) and ST-Net ($p = 6.9 \times 10^{-2}$) achieved borderline significance in the HER2+ subset (Fig. 4i). Under CV, the log-rank $p$ values along with the C-indices indicated that the predicted GE from the models together

with the RNA-Seq bulk data may not be fully prepared for clinical translation. Despite the limited performance in unseen test data, the relative strength of each method under clinical translation potential was still able to be compared and these results highlighted an area for further refinement of these models.

## Usability of methods remains a major challenge

We evaluated the user accessibility for each method through a broad survey assessing several aspects of usability. We used a scoring scheme (Fig. 5) based on existing tool quality and programming guidelines[27]. These aspects included availability of software, code quality, code assurance, documentation, behaviour, paper, reproducibility and generalisability. Overall scores for usability were obtained after summing weighted scores over each aspect, with a maximum attainable score of 31. Hist2ST achieved the highest usability score (18.8), followed by THItoGene (18.3), HisToGene (16.5), GeneCodeR (15.9), DeepSpaCE (15.5), TCGN (15.2), iStar (13.9), EGNv2 (13.8), EGNv1 (13.3), ST-Net (10.2), and then DeepPT (2, lowest due to not having any code available at time of assessment). The mean usability score was 15.1 (sd = 2.6) excluding DeepPT. The majority of methods had not been tested for robustness in their paper and we were not able to run without significant adjustments to the code. In contrast, Hist2ST excelled in these areas. Along with Hist2ST, HisToGene, TCGN and THItoGene also provided a relatively user-friendly tutorial. Aside from DeepPT, ST-Net had the lowest overall score due to issues with code execution and undocumented functions, necessitating our own implementation for benchmarking. While the scoring scheme used was able to inform which methods were the most user-accessible methods and provide a perspective on how these methods compare to exemplary tools and programmes. It is important to note that overall, there was a clear gap between the maximum usability score and the highest scoring method.

We unpacked and examined the subcategories within the usability scores that are most needed for improvement. Figure 5 shows a heatmap that highlights subcategories of usability, with pink indicating weakness and blue indicating strength. The weakest usability subcategory was code assurance, which includes unit testing, support, development model and continuous integration. Whilst most methods were available on GitHub and had a support ticketing system, only GeneCodeR and DeepSpaCE created separate releases and branches for development and master code. All methods failed to incorporate any unit testing for the prevention of unexpected code behaviour with any updates. Key strengths included open-source code, coding consistency and exposed important parameters. Subcategories that require substantial improvement (average score = 0) included the need for domain expertise to run, unexpected warnings during compilation, undocumented function outputs, limited tutorials, absence of dummy-proofing and often requiring users to write new code to apply models. Addressing these challenges would enhance overall usability, expanding the methods' user base and research potential.

## Discussion

We presented a comprehensive benchmarking study of eleven developed methods for the prediction of SGE from H&E histology images. Our evaluation extends beyond standard performance measures such as correlation and includes assessments across different spatial resolutions and tissue types. We tested the translational potential for these methods, examining model robustness to input quality, generalizability and the effectiveness of predicted genes by assessing their capabilities in survival prediction. Our study impartially evaluates the current state of methods that predict SGE from H&E histology images, with a particular focus on their potential to be applied in clinical contexts. We have also summarised the advantages and limitations of each method, and conducted a concise comparison of these methods based on benchmarking results to provide a clear user guidance in method selection for different criteria shown in the 'User Guidance' column, and proposed improvement directions from the current methods in Supplementary Table 2.

All evaluations to date strongly relied on correlation between observed and predicted gene expression as a quantitative measure to provide evidence of predictability of SGE from H&E. Our current benchmark prompts a discussion on whether correlation based on all genes should be the primary metric for evaluating methods. Most studies, including this benchmark, demonstrate that all methods consistently exhibited average correlations below 0.25 for ST data, with a typical variability of around 0.2. It is important to note that we should not anticipate any correlation between two sets of white noise data, therefore genes with no expression, or expression at the limits of detection, should not have a correlation between predicted and measured intensity. Hence, the proportion of low-expressed or non-expression genes is high with the calculation of average masking the identification of biologically meaningful concordance. As an alternative, we believe it is more meaningful to focus on HVGs and SVGs, acknowledging that correlation is meaningful only among signals and not at all for noise and consequently, our change in focus in this benchmarking study provides a more meaningful assessment of performance between methods.

Our overall observation is that the most complex deep-learning architectures involving more components were not superior in either SGE prediction or in translational potential categories. This is surprising given the promise of newer deep-learning techniques that have consistently demonstrated strong performance in other fields[28–31]. Upon closer examination of the deep-learning architectures, we found that methods effectively capturing both local and global features within single-spot patches, as exemplified by DeepPT, outperform methods attempting to incorporate additional spatial information from neighbouring spots. This observation suggests that SGE patterns in adjacent spots may not necessarily be strongly or consistently correlated. This may not be too surprising given that the spots in our evaluation data are around 150–200 μm and 100 μm apart from centre-to-centre for the ST and Visium data respectively. Furthermore, including global information from the entire image slide may introduce noise since there is a large number of genes and some genes are not expressed across all spots. This is particularly true for tissues with low heterogeneity, where global information tends to be less useful in images with smaller variations between patches. Moreover, the use of exemplars appears effective in mitigating global noise through the guidance of the most similar image patches and their GE from a reference dataset. Interestingly, transformer-based methods demonstrated better risk group stratification within HER2+ cancer subtype, suggesting that their ability to capture various histology characteristics could be more closely related to downstream clinical outcomes than SGE patterns. Therefore, it is essential to delve deeper into the biological correspondence between H&E images, GE and corresponding clinical outcomes.

When examining the translational capacity of current methods, we observed that when these methods were directly applied to TCGA data, they demonstrated in general higher patient-level correlations predicting bulk GE. Under CV the distinction between high-risk and low-risk groups was borderline statistically significant for three methods DeepSpaCE, Hist2ST and ST-Net. This suggestive discrimination of risk groups was also observed with KM curves across selected methods, providing evidence of clinical utility but demonstrating that there is room for improvement. One possible direction is to examine the suitability of deep-learning architectures that take spatial information for survival prediction, as the current approach only considers pseudobulk gene expression. It is important to note that the power of SGE extends beyond survival prediction, it enables the use of spatial information to further understand disease mechanisms. For instance, it provides the potential to perform multi-omics exploration of TCGA at a spatially resolved scale to better delineate cancer subtypes. The comparison of survival prediction performance, which existing methods have not explicitly explored, can serve as a useful measure for guiding future method development.

Based on our findings, we summarised several challenges and potential directions for future researchers. Firstly, compounding the

diversity and complexity of SGE patterns, patient effects and batch effects may contribute to the quality of various H&E images and impact the translational abilities of the methods. Currently, none of the existing methods incorporate H&E QC with predictions, thereby neglecting the importance of stain normalisation and image pre-processing. Secondly, we have shown that the SGE prediction methods perform better on HVGs and SVGs. Further investigation is needed to identify properties of genes that may make them more conducive to prediction with these methods, perhaps with a greater focus on predicting specific genes. Additionally, some methods struggle to handle densely expressed or small gene sets as they experience convergence difficulties or overfitting during training. Therefore, it is crucial to carefully select gene sets for training and prediction, combining biologically meaningful genes while balancing their sparsity to achieve optimal results. Moreover, improving the transferability or flexibility between different resolutions and cancer subtypes is essential to align with real-world applications. Using foundation models trained on a diverse range of tissues could help bridge this gap. Finally, none of the methods assessed in this study are able to fully make use of the sub-cellular resolution provided by newer platforms such as 10x Xenium[32]. As these new technologies become more prevalent, future evaluation studies will be able to test the adaptability of these approaches to be applied to higher-resolution data.

Our benchmark elucidated the comparative results of the SGE prediction methods by employing a broad set of evaluation metrics. These results uncovered the importance of accounting for SGE attributes, histology image quality, as well as the choice of deep-learning architecture in SGE prediction performance. Our findings demonstrated promising practical applications of these methods in clinical contexts while also highlighting areas for further development, particularly in predicting patient survival using spatial information. Additionally, with the rapid development of subcellular spatial transcriptomics (SST), our evaluation framework could easily be adapted and applied to new methods designed based on SST technology. We believe that this evaluation can provide a holistic, comprehensive framework for evaluating developed methods and provide guidance for future developments in this continually evolving field. Continued advancements in methods for predicting SGE from H&E can create opportunities for profound insights into complex spatial relationships and the potential to revolutionise disease risk diagnosis for patients.

## Methods

### Benchmarked methods selection criteria

We considered all methods published and in pre-print that predicted SRT data using histology images up until January 2024 (Supplementary Table 1). As there were several methods which were recently developed, not all methods included publicly available code at the time of investigation, those aren't able to be implemented based on the original papers (e.g. BrST-Net, NSL[33], TransformerST[34] and STimage[35]) were excluded as a result. Methods that didn't provide clear instruction on handling new datasets and hard to execute were excluded such as BLEEP[36] and SEPAL[37]. Methods that performed multi-model on H&E (e.g. Cottrazm[38], starfysh[39] and xfuse[40]) were also excluded as we were only investigating the applicability of H&E to ST. In addition, methods predicting or incorporating scRNA-Seq (e.g. SCHAF[41], St2cell[42] and the method proposed by Na et al.[43]) or bulk RNA-Seq (e.g. HE2RNA[44] and ISG[45]) were excluded. Although we excluded all methods predicting bulk gene expression from histology, we adapted the architecture from DeepPT for predicting ST. This was done as the model architecture was commonly used as a backbone in the deep-learning field and it's relatively simple to reproduce, and the authors had also aimed to predict clinical outcomes using predicted SGE from their method, which was one of our evaluation categories of interest.

### Datasets and pre-processing

The same publicly available ST data that were employed in previous studies were used for benchmarking, along with publicly available 10x Visium data. In addition, H&E images matched with bulk gene expression profiles from TCGA were also used.

**Human HER2-positive breast tumour (HER2+ ST)[46].** The HER2+ dataset was measured using ST to investigate SGE in HER2-positive breast tumours, from which the original investigators discovered shared gene signatures for immune and tumour processes. The dataset consists of 36 samples of HER2-positive breast tissue sections from 8 patients, with tissue annotations available for one slide per patient.

For the histology images, 224 × 224 pixel patches were extracted around each sequencing spot (112 × 112 pixel patches were used for TCGN and THItoGene due to optimised model settings). For the SGE data of each tissue section, the top 1000 HVGs for each section were considered and those genes that were expressed in less than 1000 spots across all tissue sections were excluded. This resulted in 785 genes for the training of all methods.

**Human cutaneous squamous cell carcinoma (cSCC ST)[47].** The cSCC dataset was measured using ST to understand cellular composition and architecture of cSCC. The original investigators discovered shared gene signatures for immune and tumour processes. Whilst the original dataset contained single-cell RNA sequencing along with ST and multiplexed ion beam imaging, we analysed only the ST data. This ST data consists of 12 samples from 4 patients.

For the histology images, 224 × 224 pixel patches were extracted around each sequencing spot (112 × 112 patches extracted for TCGN and THItoGene). For the gene expression data of each tissue section, the top 1000 HVGs for each section were considered and those genes that were expressed in less than 1000 spots across all tissue sections were excluded. This resulted in 997 genes for the training of all methods.

**TCGA breast invasive carcinoma (TCGA-BRCA).** Paired RNA-Seq bulk gene expression and H&E images from TCGA Network were used to evaluate model generalisability. This dataset was chosen as the predictions on these H&E images could be evaluated through the matched gene expression, and since the tissue type is the same as the data the models were trained on (HER2+ ST).

TCGA-BRCA data was downloaded using the TCGAbiolinks package[48] version 2.29.6. RNA-Seq data was obtained by following query options: project = 'TCGA-BRCA', data.category = 'Transcriptome Profiling', data.type = 'Gene Expression Quantification', experimental.strategy = 'RNA-Seq' and workflow.type = 'STAR - Counts'. Histology images were obtained by the following query using project = 'TCGA-BRCA', data.category = 'Biospecimen', data.type = 'Slide Image' and experimental.strategy = 'Diagnostic Slide'. We only considered images that had one associated RNA-Seq entry and were either stage I, stage III, or stage IIA due to storage limitations. Overall, we considered 671 TCGA images. Images were read using the Open-Slide package and converted to jpg from svs. Images were then split into 224 × 224 pixel patches and filtered out if the sum of the RGB values was greater (or more white) than a pure RGB (220, 220, 220) image. Further clinical information containing variables to define breast cancer subtypes for samples was downloaded from Broad GDAC Firehose.

For Hist2ST, the model fuses image patch embedding with spot coordinate embedding; a few TCGA images have more than 2048 spots exceeding the number of embedding dimensions. Therefore, the selected TCGA images for Hist2ST were cropped into 9 large patches before further splitting into 224 × 224 patches.

**10x Visium HER2-positive breast cancer and Hercep Test 2+ breast cancer.** The two Visium breast cancer datasets were obtained

from the 10x Genomics platform, measuring breast tissue from BioIVT Asterand. H&E images were captured using a 20× objective. The first dataset Visium-HER2+ includes two slides from an ER-positive, PR-negative, HER2-positive Invasive Ductal Carcinoma patient, while the second dataset Visium-Hercep-Test2+ contains one slide from an ER-positive, PR-negative, Hercep Test 2+ Invasive Ductal Carcinoma patient. Despite both datasets being generated using the 10x platform, they exhibit slight differences in spatial resolution.

For the histology images, 224 × 224 pixel patches were extracted around each sequencing spot (112 × 112 patches extracted for TCGN and THItoGene). In terms of gene expression data, the top 1000 HVGs from each tissue section were selected, and genes expressed in fewer than 1000 spots across all sections were excluded. This resulted in 990 genes being used for model training.

**10x Visium human healthy kidney.** The Visium-Kidney dataset, provided by Avaden Biosciences and generated using the 10x Genomics platform, includes one kidney healthy tissue from a single patient. The corresponding H&E image was captured at 20× magnification.

For the histology images, 224 × 224 pixel patches were extracted around each sequencing spot (112 × 112 patches extracted for TCGN and THItoGene). In terms of gene expression data, the top 1000 HVGs of the whole slide were considered and those genes that were expressed in less than 1000 spots were excluded. This resulted in 992 genes for the training of all methods.

### Settings used and practical implementation of existing methods

**GeneCodeR.** GeneCodeR was the only method that did not use deep learning for its model. The generative encoder was set up as the following relation between the histology images ($H$) and SGE values ($E$):

$$\alpha_{H_{k,i}} H_{i,l} \beta_{H_{l,c}} = \alpha_{E_{k,i}} E_{i,j} \beta_{E_{j,c}} \qquad (1)$$

where $\alpha$ and $\beta$ are matrices that transform the sample and features across the datasets into the same space, $k$ and $c$ are the dimensions of the reduced dimensional encoding for the samples and features respectively, $i$ is the number of samples, $j$ is the number of genes and $l$ is the number image dimensions.

The following hyperparameters were used for the coordinate descent updating algorithm: the dimensions of the encoding space for the samples (i_dim) of $k = 500$, the dimensions of the encoding space for the features (j_dim) of $c = 500$, initialisation for the alpha_sample = 'rnorm', initialisation for the beta_sample = 'irlba', max iterations of 20, tolerance of 5, learning rate of 0.1 and batch size of 500.

**HisToGene.** HisToGene, is a deep-learning method for SGE prediction from histology images. HisToGene employs a vision transformer (ViT), a state-of-the-art method for image recognition to predict SGE and can also be used to enhance the resolution of SGE. The SGE values used for training were the normalised unique molecular identifier (UMI) counts from each spot. The normalised counts were derived by taking the UMI count for each gene divided by the total UMI counts across all genes in that spot, multiplied by 1,000,000 and then transformed to a natural-log scale.

The HisToGene model was implemented using PyTorch with the following hyper-parameters: a learning rate of $10^{-5}$, the number of training epochs of 100, a dropout ratio of 0.1, 8 Multi-Head Attention layers and 16 attention heads. The final model used was the model after all epochs, as per the HisToGene pipeline.

**DeepSpaCE.** The Deep-learning method for spatial gene clusters and expression, DeepSpaCE, was developed for the prediction of tissue sections and for the enhancement of resolution of SGE. A VGG16 deep CNN model architecture is used for SGE prediction and semi-supervised training is available in DeepSpaCE. The DeepSpaCE

pipeline was originally developed to read data output from the Space Ranger pipeline (10x Genomics), so the code had to be configured to read the ST data. The SGE data were first normalised using the regularised negative binomial regression method implemented in the SCTransform function from Seurat (v3.1.4) R package. The data used for training the method was the corrected UMI returned by the vst function from the sctransform package with the max and min values clipped to $\left( -\sqrt{n_{spots}/30}, \sqrt{n_{spots}/30} \right)$.

The DeepSpaCE model was then implemented using the VGG16 model as the backbone with the following hyper-parameters: maximum number of training epochs of 50 (with training stopping before 50 if there is no improvement within 5 epochs), a learning rate of $10^{-4}$ and a weight decay of $10^{-4}$. The final model used was the model with the highest correlation in the validation set, as per the DeepSpaCE pipeline.

**Hist2ST.** Hist2ST is a deep-learning method predicting SGE from histology images comprising three main frameworks: Convmixer, Transformer and GNN. It takes histology image patches, each sized 224 × 224, as inputs to the Convmixer framework, extracting 2D vision features from each patch. These features capture important local and global information within the patch. The embedded patch features are then fused with the corresponding embedded spot coordinates to create an enriched representation. These combined features are subsequently fed into the transformer framework, which considers the spatial relationships among the spots. Finally, the GNN module learns the relationships among neighbouring patches to enhance local spatial dependencies from the entire image. To improve model generalisability and performance, zero-inflated negative binomial distribution (ZINB) and self-distillation models were applied after the GNN. This allowed for capturing the probability of observing zero counts separately from the distribution of non-zero counts. Additionally, to mitigate the impact caused by small ST data by fusing the features between the anchor image patch and the augmented image patches. The gene counts for each spot were divided by the total counts of that spot and multiplied by $10^6$. Then natural-log transformation $\log(1 + x)$ was applied to the normalised count $x$.

The model was implemented using Python and PyTorch and ran with the default parameters stated in the original paper. These included: batch size of 1, learning rate of $10^{-5}$, 350 epochs of training and applying random grayscale, rotation and horizontal flip to image patches for self-distillation strategy. Some hyperparameter changes were made for benchmarking including increasing the image patch size from 112 × 112 to 224 × 224 and reducing the number of input and output channels in the depthwise and pointwise convolutional blocks from 32 to 16 due to limited computational power. Image embedding dimension was increased from 1024 to 2048 as the original code of setting embedding size was hard coded based on the image patch size and the number of channels. The model was trained using a sum of MSE loss of predicted and ground truth expression levels, ZINB and Distillation mechanisms.

**ST-Net.** ST-Net was one of the earliest methods proposed to infer ST from H&E images. ST-Net leveraged a CNN to capture gene expression heterogeneity in histology. A DenseNet-121 backbone was used to extract a 1024-dim embedding of image patches (224 × 224 pixels in size), then a fully connected layer (with the same number of outputs as the number of genes) was applied to predict gene expressions of a spot. The weights of all convolutional layers were pretrained on ImageNet. The authors originally used this architecture to predict the expressions of 250 genes.

ST-Net was implemented in Python and PyTorch using hyper-parameter values as outlined in the original paper. These included 224 × 224 pixel-sized input image patches, batch size of 32, learning rate of $10^{-5}$, 50 epochs of training and applying image rotations and

flips during training. The DenseNet-121 backbone used was pretrained on ImageNet. ST-Net was trained using the MSE loss function, computed between predicted and ground truth expression levels.

**DeepPT**. The DeepPT architecture consisted of three main components in the following order: a CNN, autoencoder and a multi-layer perceptron (MLP). The CNN (ResNet−50 that was pretrained on ImageNet) was used to extract features from image tiles and produced a 2048-dim embedding. The autoencoder comprised an input, hidden and output layer that compressed the image embedding into a 512-dim vector. This step was designed to reduce data sparsity, avoid overfitting and reduce noise. The compressed vector was then fed into a 3-layer MLP (consisting of an input, hidden and output layer) to predict SGE. The authors treated each gene as an individual task in a multi-task paradigm; hence DeepPT can learn shared features between different genes.

DeepPT was implemented in Python and PyTorch using hyperparameter values as outlined in the original paper. These included: training for a maximum of 500 epochs, learning rate of $10^{-3}$, batch size of 32, dropout set to 0.2 and applying rotations to the input images during training as data augmentation. The size of input image patches was $224 \times 224$ pixels. DeepPT was trained using the MSE loss function, computed between predicted and ground truth expression levels.

**EGNv1**. This is a method using exemplar learning to boost gene expression prediction from nearest exemplar images. The EGN framework comprises three main components: an extractor (pretrained ResNet50) to obtain image representations using unsupervised exemplar retrievals; a ViT backbone to progressively extract representations of the input image; and an Exemplar Bridging (EB) block to adaptively revise the intermediate ViT representations by using the predefined nearest exemplars. Following these components, there is a simple attention-based prediction block to complete the gene expression prediction. Log transformation and min-max normalisation was applied to the ground truth gene expression.

The model was developed in Python and PyTorch following the default settings from the original paper. The EGN component was trained from scratch over 50 epochs, with a batch size of 32. The ViT backbone featured a patch size of 32, an embedding dimension of 1024, a feedforward dimension of 4096, 16 attention heads and a depth of 8. Additionally, the EB block, integrated with the ViT backbone, operated at a frequency of 2, with 16 heads and a dimension of 64, utilising the 9 nearest exemplars. The size of the model input was modified from $256 \times 256$ to $224 \times 224$ and used a learning rate of $5 \times 10^{-5}$ with a cosine annealing scheduler and weight decay of $10^{-4}$. The model was optimised by a combination of mean squared loss and batch-wise PCC loss.

**EGNv2**. EGNv2 is a subsequent version of EGNv1. Similar to EGNv1, it also has three main components: an extractor (pretrained ResNet18) to obtain image representations using unsupervised exemplar retrievals; a graph method to connect image patch and its exemplars as a graph; a graph convolutional network to process image and exemplar features, and a graph EB block to adaptively revise the image features using its exemplars. Gene expression prediction is obtained through a simple attention-based prediction block. Log transformation and min-max normalisation was applied to the ground truth gene expression.

We implemented the model in Python and PyTorch Geometric frameworks using default hyperparameters stated in the original paper. The training was conducted for up to 50 and 300 epochs for HER2+ and cSCC ST datasets with a batch size of 1 respectively. We utilised a GraphSAGE backbone with 512 hidden dimensions spread across 4 layers and set the learning rate at $5 \times 10^{-4}$ with a cosine annealing scheduler and weight decay of $10^{-4}$. The model's input size was adjusted from $256 \times 256$ to $224 \times 224$ pixels and optimised by a combination of mean squared loss and batch-wise PCC loss.

**TCGN**. TCGN integrates convolutional layers, transformer encoders and GNN to process single-spot images. Initially, imaging features pass through convolutional layers to extract local details and transformer encoders to capture long-range dependencies within a spot. These features are then input into two GNN blocks which analyse how these graph features interact with features extracted by the CNN and transformer. Afterwards, features proceed through additional CMT blocks, which efficiently combine the convolutional and transformer layers, optimising self-attention with fewer parameters. The gene counts for each spot were divided by the total counts of that spot and multiplied by $10^6$. Then natural-log transformation $\log(1+x)$ was applied to the normalised count $x$.

The model was built using Python and PyTorch, following the default settings from the original paper. It was trained for up to 80 epochs with a learning rate of $10^{-5}$. The Adam optimiser was used, with gradient coefficients set to 0.9 and 0.999, and a batch size of 32. Specific configurations included: CMT blocks set to default parameters of CMT-Ti, a top-k of 4 in the GNN block's connection analyser and the prediction head featuring a last $1 \times 1$ convolutional layer with 1280 channels and two linear layers with 5120 and 785 dimensions, respectively. The input size was maintained at $112 \times 112$ pixels, optimised from the parameters. The model was trained using mean square error as the loss function to measure gene expression levels between predictions and the ground truth.

**THItoGene**. THItoGene is a hybrid neural network that combines dynamic convolutional and capsule networks. It starts by using the multidimensional attention mechanism of dynamic convolution to extract detailed local visual information from input images. The network then employs the Efficient-CapsNet module, which learns spatial relationships and distribution patterns among cells. The model further integrates spot location and low-level image features for global analysis through the ViT module. Finally, it builds a neighbourhood network using the Graph Attention Network (GAT) module, which adaptively characterises the relationships between spot positions and gene expression. The gene counts for each spot were divided by the total counts of that spot and multiplied by $10^6$. Then natural-log transformation $\log(1+x)$ was applied to the normalised count $x$.

The model, implemented in Python and PyTorch, followed the default settings from GitHub. It trained for up to 200 epochs with a learning rate of $10^{-5}$ and used the Adam optimiser. Key configurations included a batch size of 1, a capsule network with a routing vector dimension of 64 and 20 capsules, and the ViT and GAT modules with [16, 8] heads. Number of transformer blocks was set to 4. The size of input image patches was kept as $112 \times 112$ pixels to maintain optimised performance. The model utilised the Negative Log-Likelihood loss function to measure the accuracy of predictions against the ground truth expression levels.

**iStar**. iStar introduced a method for predicting near-single-cell level SGE from spot-based measurements. This method leverages hierarchical imaging features and super-resolution techniques to achieve fine resolution in gene expression predictions. A key component of iStar is the use of a pretrained hierarchical vision transformer (HViT), which extracts histology features at two distinct scales: $16 \times 16$ pixels to capture fine tissue details and $256 \times 256$ pixels to capture larger tissue structures. Using these features, iStar constructs a weakly supervised model to predict gene expression at the superpixel level. The method splits the gene expression measured from each spatial spot into smaller values, assigning them to individual super-pixels based on their histology features. Both the ground truth and predicted gene expression are treated as images, normalised to a range of [0, 1].

The model was implemented in Python using PyTorch and trained for 400 epochs with a learning rate of $10^{-4}$, optimised via Adam. The input images were rescaled so that each pixel corresponded to

$0.5 \times 0.5\,\mu m$, ensuring that a $16 \times 16$ pixel patch represented an $8 \times 8\,\mu m^2$ area, approximately the size of a single cell. Each $16 \times 16$ patch was embedded into a 579-dimensional space. Mean squared error (MSE) loss was used for the weakly supervised learning process. Practical implementation of iStar can be found in Supplementary Method 1.

## Performance evaluation

Figure 2a, b was created using the R package funkyheatmap. Scales of the bars in Fig. 2a are relative within each category where the maximum value for $x$-axis for the bars represents the lowest average rank (with rank 1 being the best), and the minimum value of the $x$-axis for the bars represents the highest average rank (not necessarily zero). The blank spaces in the rows indicate that the methods were not applicable to those categories. Each entry in Fig. 2a, b represents the ranking of the corresponding metric across all methods.

Metrics of predicted SGE from the test samples of HER2+ and cSCC ST datasets were used to build the 'ST Gene Expression Prediction' category, while the 'Visium Gene Expression Prediction' category was based on the prediction results from Visium-HER2+ and Visium-Kidney test samples. Methods that failed to perform were ranked lowest in the corresponding metrics. For the 'Model Generalisability' and 'Clinical Translational Impact' categories, metrics were derived from analyses using TCGA data. 'Reimplementation' refers to situations where a method had to be implemented based on the original paper because the code was not publicly available. 'Whole Image Spatial' indicates that the model learns global information from the entire H&E slide.

Overall rankings were calculated by first taking the ranks of each method in each metric. These ranks were then averaged over all metrics within the same evaluation category to obtain an overall score for each category. Then the overall rankings of each category were averaged over all categories to obtain a final overall score/rank.

## Evaluation metrics

Scale-independent metrics were chosen to compare performance from different methods since some models were trained on normalised/augmented counts rather than the raw gene counts, see Table 2. This was done as opposed to back-transforming normalised counts as it may introduce potential issues, such as loss of information or bias. This was particularly important as methods transformed the original data for specific reasons.

## Gene expression prediction

The SGE prediction of the ST datasets was evaluated using 4-fold CV. The HER2+ and cSCC ST datasets were split into four folds, ensuring that adjacent tissue sections from the same patient were grouped within the same fold. In each iteration of the CV, two folds were used as a training set, one fold was used as a validation set and the remaining fold was used as a test set. For the Visium predictions, a cross-study evaluation (referred to as Visium-HER2+) was conducted by training on the Visium-Hercep-Test2+ dataset and testing on the Visium-HER2+ dataset. The models trained using HER2+ ST (best fold) were applied to the Visium-HER2+ data for external validation (referred to as ST-Visium-HER2+). For the Visium-Kidney dataset, the tissue was evenly divided into four non-overlapping sections, and 4-fold CV was performed following the same settings as for the ST datasets. The training set was used to train the parameters of the model. The validation set was used to evaluate models during training to allow parameters to be tuned. The best model within each fold was then chosen based on the epoch that produced the best PCC in the validation set. Results presented in the paper were calculated using the predictions from the best performing model (using the test set) out of the four folds of the CV, unless specified otherwise) The same CV splits were used for each model to ensure fair comparison. Metrics under this category of

evaluation were first calculated at an image level (e.g. correlation was measured for each gene per image), and then averaged over each patient (this is referred to as average correlation of specific genes), then averaged over each gene.

For the clustering in Fig. 3e, the default clustering in the Heatmap function from the ComplexHeatmap R package[49] was used. More specifically, the heatmap utilised average-linkage hierarchical clustering over the Euclidean distances, with the hierarchy reordered through the reorder.dendrogram function from the stats R package[50].

HVGs and SVGs in Fig. 3a, f, g were deduced from the ground truth SGE from the ST and Visium datasets. The modelGeneVar function, followed by the getTopHVGs function with prop = 0.1 from the scran R package[51] was used to obtain the HVGs in each dataset. For SVGs selection, genes with adjusted $p < 0.05$ were identified as SVGs in each image sample in each dataset using SPARK-X package[52] in R. For HER2+ ST, only the SVGs that appear in more than 30 out of 36 samples were selected and correlation was calculated based on top 20 genes ranked by adjusted $p$ value. For cSCC ST, a common set of top 20 SVGs of each sample from HER2+ ST were selected and then further narrowed down by top 20 genes sample-wise by adjusted $p$ value. For Visium-HER2+, the top 20 SVGs shared between the two slides were selected. For Visium-Kidney, the top 20 SVGs across all spots were chosen, as only a single H&E image was available.

The sparsity of the gene expression matrix was determined by calculating the proportion of zeros for each gene across all spots. Genes with a sparsity greater than 0.6 in both the training and test sets were selected, resulting in 145 genes for Visium-Kidney and 274 genes for Visium-HER2+. This selection was used to investigate the impact of different levels of gene matrix sparsity.

## Model generalisability

For TCGA images, a pseudobulk expression for each image was calculated by predicting the SGE for each image and then averaging the gene expression across each image tile. Pseudobulk values were then transformed as follows:

$$x' = \log(\max(0, x) + 1) \qquad (2)$$

where $x$ is the original predicted pseudobulk values for each gene and image.

The pseudobulk gene expression was then evaluated against the true bulk gene expression associated with the H&E image from the TCGA database. For the patient-level correlations, only the log-transformed gene values that were greater than 5 were included. This is appropriate to filter out noise for a more informed evaluation.

Histology QC metrics were calculated using HistoQC python package. Whole slide images were converted to single-file pyramidal tiled TIFF format using nip2 interface for the libvips[53] package. Then HistoQC python package was run on each of the converted images for stage I breast cancer samples only. For Fig. 4a, the first principal component from a PCA using colour-related metrics was plotted against the patient-level correlation between TCGA RNA-Seq and predicted pseudobulk GE. Colour-related metrics included: rms_contrast (the standard deviation of the pixel intensities across the pixels of interests), michelson_contrast (measurement of image contrast defined by luminance difference over average luminance), grayscale_brightness (mean pixel intensity of the image after converting the image to grayscale), chan1_brightness, chan2_brightness, chan3_brightness (mean pixel intensity of the image of the red, green and blue colour channels respectively), chan1_brightness_YUV, chan2_brightness_YUV and chan3_brightness_YUV (mean channel brightness of red, green and blue colour channels of image after converting to YUV colour space, respectively).

## Clinical and/or translational impact

Out of the curated TCGA images, we split samples according to breast cancer (BC) subtypes as HER2+ BC, TNBC and luminal BC. The clinical information such as ER-, PR- and HER2- were defined according to the oestrogen, progesterone and HER2 receptor status variables and these were used to define BC subsets.

For each BC subset, we used the RNA-Seq to build a survival model using only the genes present in the HER2+ ST dataset. Alongside the model built from RNA-Seq, we used the predicted pseudobulk GE from each of the benchmarked methods to build additional survival models for comparison. Pseudobulk values were normalised through mean centring (Supplementary Figs. 17 and 18). Days to death and days to last follow-up (where there was no death) were converted to years and then used as the survival times, and death used as the survival outcome. We used the top five genes to build survival models within each BC subset and GE data. These top genes were determined by the C-indices of univariate cox models for each gene. A multivariate cox regression was then built using these top five genes.

Each model was assessed through the calculation of C-index and a log-rank $p$ value for each BC subset and GE data (bulk RNA-Seq GE and predicted pseudobulk GE). C-Index was calculated by comparing the ground truth event times and events with model predictions for each patient. Predictions from the cox regression models were then used to categorise patients into high-risk and low-risk, using the median prediction to split patients. KM curves were then constructed within high-risk and low-risk. The log-rank test was then used to obtain a $p$ value to quantify the difference between the survival of the two categories.

Two types of evaluation were used for survival models within each BC subset and GE data. We calculated C-indices and log-rank $p$ values based on: (i) methods described in Xu et al. which is equivalent to a resubstitution model and (ii) a CV framework. For (i), models were built using all patients and then predictions for these patients were used for the calculation of C-index and log-rank $p$ value. For (ii), we used a 3-fold CV with 100 repeats. This meant that patients were randomly split into three subsets. Each subset was used as a test set of patients while the rest of the patients were used to train models. By repeating this process 100 times, we measured the C-indices for each test set to obtain an average C-index over 300 values. We also obtained the average prediction for each patient over all repetitions which was then used to categorise patients into high and low-risk to calculate a log-rank $p$ value.

## Usability

The user accessibility data was generated extending the scoring scheme to assess usability of trajectory inference methods. This scoring scheme is based on existing tool quality and programming guidelines found in the literature. In addition to this scheme, we added two sections: (i) reproducibility, which assesses whether we were able to run the analysis pipeline with minimal changes to code; and (ii) generalisability, which assesses whether methods included functionality to preprocess and predict on new images. These were important categories addressing whether users would be able to run the inference methods under a clinical translational setting. Additionally, we applied a slightly higher weight to these categories as part of scoring methods.

## Performance

For calculating performance, memory and time taken to train one image from the HER2+ ST dataset (sample A1) over 10 epochs was used as a unit for comparison across methods (Supplementary Fig. 19). We achieved this by averaging the values across all epochs and then taking the ranks. Peak and instantaneous memory was calculated via the get_traced_memory from the tracemalloc module for all python methods whilst the peakRAM function from the peakRAM library was used for GeneCodeR and R method. The performance of EGNv1 and EGNv2 were calculated based on their main network, additionally with the average time of corresponding exemplar retrieval and graph construction (EGNv2 only) over 50 epochs, which is the number of epochs they were trained on since the exemplars and graphs only need to be built once for the whole training process. Similar calculations were applied to iStar, where the time for extracting image embeddings was averaged over 400 epochs and added to the main model training time of one epoch.

All results reported in this section were obtained using Linux with Intel(R) Xeon(R) Gold 6338 CPU @ 2.00 GHz and 1TB memory. Additionally, four NVIDIA RTX A5000 GPUs with 24GB of memory each were available for GPU-based methods.

All analysis of model performance was performed using R version 4.3.1. Visualisations not already referenced were generated using ggplot2[54].

## Reporting summary

Further information on research design is available in the Nature Portfolio Reporting Summary linked to this article.

## Data availability

All datasets used in this study are publicly available and were downloaded from the following links: (1) human HER2-positive breast cancer ST data reported in Anderson et al. (https://github.com/almaan/her2st); (2) Human cSCC reported in Ji et al. (https://www.ncbi.nlm.nih.gov/geo/query/acc.cgi?acc=GSE144240); (3) TCGA-BRCA data was requested using the TCGAbiolinks package, clinical information from Broad GDAC Firehose (http://gdac.broadinstitute.org/); (4) 10x Visium Human Breast Cancer (Visium-HER2+) Block A Section 1 and Section 2 were downloaded from (https://www.10xgenomics.com/datasets/human-breast-cancer-block-a-section-1-1-standard-1-1-0) and (https://www.10xgenomics.com/datasets/human-breast-cancer-block-a-section-2-1-standard-1-1-0); (5) 10x Visium Whole Transcriptome Human Breast Cancer (Visium-Hercep-Test2+) was downloaded from (https://www.10xgenomics.com/datasets/human-breast-cancer-visium-fresh-frozen-whole-transcriptome-1-standard); (6) The 10x Visium Human Healthy Kidney (Visium-Kidney) dataset was downloaded from (https://www.10xgenomics.com/datasets/human-kidney-11-mm-capture-area-ffpe-2-standard). Source data are provided with this paper.

## Code availability

The code used to develop the benchmark pipeline and generate results in this study is publicly available and has been deposited in GitHub at https://github.com/SydneyBioX/HEtoSGEBench, under Apache-2.0 license. The specific version of the code and data associated with this publication is archived in Zenodo and is accessible via https://doi.org/10.5281/zenodo.14602489[55].

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

## Acknowledgements
The authors thank all their colleagues, particularly at the Sydney Precision Data Science Centre, Charles Perkins Centre and Biomedical Data Analysis and Visualisation Lab for their support and intellectual engagement. Special thanks to Yingxin Lin, Yue Cao, Lijia Yu, Andy Tran, Hao Wang, Andrew Sawyer and Yunwei Zhang for their contributions in weekly discussions. This work is supported by the AIR@innoHK programme of the Innovation and Technology Commission of Hong Kong to J.Y., J.K., E.P., X.F. The work is also supported by Judith and David Coffey funding to J.Y.; NHMRC Investigator APP2017023 to J.Y. and C.W. Australian Research Council Discovery project (DP200103748) to J.K. and C.W.; Discovery Early Career Researcher Awards (DE220100964) to S.G. and (DE200100944) to E.P. Chan Zuckerberg Initiative Single Cell Biology Data Insights grant (2022-249319) to S.G.; and USyd-Cornell Partnership Collaboration Awards to S.G. Research Training Program Stipend Scholarship to A.C. The funding source had no role in the study design, in the collection, analysis, and interpretation of data, in the writing of the manuscript, or in the decision to submit the manuscript for publication.

## Author contributions
J.Y. and E.P. conceived and led the study with design input from S.G. and J.K. C.W. and A.C. co-led the development of the benchmarking framework input and guidance from J.Y. and E.P. C.W., A.C. and X.F. shared the data curation and processing implementation of all existing methods and ran the benchmarking studies. C.W. and A.C. co-led the development and interpretation of the evaluation framework with input from J.Y., E.P., S.G., J.K. and X.F. All authors contributed to the writing, editing, and approval of the manuscript.

## Competing interests
The authors declare no competing interests.
