## [Transparent Peer Review file · Nature Communications]

Benchmarking the translational potential of spatial gene expression prediction from histology

Corresponding Author: Professor Jean Yang

Version 1:

Reviewer comments:

Reviewer #1

(Remarks to the Author)

In this manuscript, Wang et al. have conducted a comprehensive benchmarking study of methods for predicting spatial gene expression from histological images. The authors evaluated various computational approaches that integrate spatial transcriptomics data, aiming to enhance the understanding of how histological images can inform about gene expression patterns across different tissue types. I commend Wang and colleagues for their rigorous approach to evaluating the performance of diverse methodologies in predicting spatial gene expression from histology images.

Major Comments:

1. The datasets employed in this manuscript were generated using low-resolution ST technologies. I recommend utilizing technologies with various spatial resolutions, such as stereo-seq or slide-seq, to enhance the objectivity of the evaluations. The performance of different spatial technologies could significantly influence method selection, which is crucial for biologists and developers striving to refine these techniques.
2. While the manuscript assesses the accuracy of methods by measuring gene expression correlations within tissues, this approach may not be comprehensive enough. Considering the datasets include gold-standard spatial delineations in pathological images, I suggest incorporating additional metrics to enhance the thoroughness of the evaluations.
3. The manuscript should discuss the limitations of the current methods in detail and demonstrate how integrating recommended new approaches with existing tools can address these issues.
4. The study evaluates method performance solely on tumor tissue sections. Given the significant heterogeneity of tumor tissues, this might not adequately reflect the methods' effectiveness. I recommend conducting evaluations across a broader range of tissue types.
5. It would be beneficial to assess the precision of methods when handling data with varying degrees of gene expression matrix sparsity, spatial resolution, and gene counts to thoroughly evaluate their performance.
6. In the section "Characteristics of methods play a large role in gene-specific performance," the authors have not discussed which neural network architectures are best suited for addressing this issue. This section more closely resembles an ablation study and lacks a detailed analysis. Additionally, from Figure 3c, I am unable to obtain sufficient information to assess the similarity in performance clustering among methods.

Minor Comments:

1. Please verify the necessity of redeclaring '(Figure 1b)' at line 114 to avoid potential redundancy.
2. The statement at line 150, 'GNAS (r = 0.47) and FASN (r = 0.47),' appears inconsistent, as the correlation coefficients for these two genes by EGNv2 seem not to be the same according to Figure 2e. Please verify this data.
3. At line 155, 'MYL12B, with an average correlation of 0.25,' it is essential to clarify whether 'average' refers to the mean correlation across multiple samples or tests, and how it was calculated.

4. Review the use of abbreviations for professional terms throughout the manuscript. Ensure terms like 'GNN' are introduced with both the full name and abbreviation on first mention and used consistently thereafter, avoiding repeated annotation of both the full name and abbreviation at lines 66 and 492.

Reviewer #2

(Remarks to the Author)

In this paper, the authors benchmark multiple methods for predicting spatial gene expression from histology data. Diverse metrics are used for evaluation, including not only prediction accuracy but also usability and computational efficiency. The paper provides valuable insights into the performance of the prediction models. However, there are areas that need major improvements.

1. For experiments on the HER2+ and CSCC datasets, it is unclear whether any subject appears in both the training and the testing sets, since each subject can contain multiple adjacent tissue sections. For the TCGA dataset, are training and testing samples both from TCGA, or are training and testing performed on different studies?
2. For predicting spatial gene expression from histology images, an important aspect is cross-subject and cross-study prediction, but this type of prediction seems to be not rigorously evaluated in this paper. For the TCGA dataset, evaluation is only performed at the bulk-tissue level, which does not measure the prediction accuracy of spatial gene expression. For the HER2+ and CSCC datasets, it is not clear whether the authors have performed cross-study evaluation. Even if so, the experiments are still limited to two datasets. Including only 3 datasets is a major limitation for a benchmarking paper. Given the public availability of numerous spatial transcriptomic datasets, it would greatly improve this paper's contribution to spatial omic research if multiple independent datasets are included and evaluation is performed by using some datasets for training while others for testing.
3. Fig 2a-b: It is useful to summarize the evaluations of all methods across various metrics in one chart, but it is unclear whether each entry of the table is a subjective score or computational evaluation. Overall, the numerical rigor of Fig2a-b needs to be improved.
4. Fig 2a-b: Which datasets are used for evaluating the computational efficiency of each method?
5. For each panel in Figure 2, which dataset is used for evaluation? It is also not clearly indicated whether training and testing are performed in the same subject/study or across different subjects/studies.
6. Fig 2b: This is a very useful chart, and visualizing ranks is also a good approach. However, the legends are not very visually distinguishable, since ranks are represented by the shape and size of the markers. It would be easier for readers to find the best method for each column if the ranks are color-coded (e.g. by using a heat map).
7. Fig 2a, third column: What does "reimplemented" mean here?
8. Fig 2d: What does each point in the box plot represent? A gene?
9. Fig 3e: What does the heat map look like if the genes are sorted by mean or variance?
10. If prediction is performed for each spot, how SSIM is computed for the predicted and ground truth spatial gene expression, since standard SSIM is only applicable to pixelized data?
11. I might have missed this but how are SVG and HVG defined

Reviewer #3

(Remarks to the Author)

The authors benchmarked several methods for predicting gene expression data from histological images. Overall, the insights and conclusions are too general to merit publication in Nature Communications. Additionally, the authors used only a few tumor types or samples for benchmarking and did not consider the recent surge in new papers in this field, such as iStar 1.

1 Zhang, D. et al. Inferring super-resolution tissue architecture by integrating spatial transcriptomics with histology. Nature Biotechnology, doi:10.1038/s41587-023-02019-9 (2024).

Version 2:

Reviewer comments:

Reviewer #1

(Remarks to the Author)

The authors revised all my concerns.

(Remarks on code availability)

Reviewer #2

(Remarks to the Author)

While the authors have addressed most my comments, my biggest concern remains that only a few SRT datasets are

included in this study. Even with the newly added experiments, there are only 5 SRT datasets in total. By contrast, there were at least a dozen of Visium datasets (including H&E images) on 10x Genomics' website alone one year ago, and half a year ago two more publicly sourced multi-study SRT meta-datasets were curated (HEST-1k and STimage-1K4M), each containing about 1000 SRT+H&E slides. Extending the authors' current study setup for Visium to more datasets, which have the same formats, should be straightforward. Given that generalizability is a key bottleneck in current algorithms for predicting SRT from histology, more datasets need to be included to make the results more generalizable to meet the standard of Nature Communications.

(Remarks on code availability)

REVIEWER COMMENTS

Reviewer #1 (Remarks to the Author):

In this manuscript, Wang et al. have conducted a comprehensive benchmarking study of methods for predicting spatial gene expression from histological images. The authors evaluated various computational approaches that integrate spatial transcriptomics data, aiming to enhance the understanding of how histological images can inform about gene expression patterns across different tissue types. I commend Wang and colleagues for their rigorous approach to evaluating the performance of diverse methodologies in predicting spatial gene expression from histology images.

Major Comments:

1. The datasets employed in this manuscript were generated using low-resolution ST technologies. I recommend utilizing technologies with various spatial resolutions, such as stereo-seq or slide-seq, to enhance the objectivity of the evaluations. The performance of different spatial technologies could significantly influence method selection, which is crucial for biologists and developers striving to refine these techniques.

Response:

We appreciate the reviewer's suggestion to include various spatial resolutions. Despite having some components of the data being publicly available, Stereo-seq or Slide-seq often have no corresponding H&E images available. Therefore, we have instead focused on exploring different spatial resolutions using 10x Visium data.

In our initial submission, we demonstrated the performance of spatial transcriptomics (ST) data using:

- the HER2-positive breast tumour dataset (HER2+ ST), with a spatial resolution of 100 μm and a centre-to-centre distance of 200 μm between spots; and
- the cutaneous squamous cell carcinoma dataset (cSCC ST), with a spatial resolution of 110 μm and a centre-to-centre distance of 150 μm .

We have now included results based on three additional higher resolution datasets: the 10x Visium HER2-positive Breast Cancer (Visium-HER2+) and Hercep Test 2+ Breast Cancer (Visium-Hercep-Test2+) as well as the Human Healthy Kidney (Visium-Kidney) datasets. These datasets have a spatial resolution of 55 μm and a centre-to-centre distance of 100 μm . Details about the additional datasets and corresponding experiments settings have been added in Material and Methods under "Datasets and pre-processing" and "Gene expression prediction", respectively.

Based on the new experiments utilising high-resolution datasets, we presented a new results section named "Spatial resolutions and gene matrix sparsity influence methods' performance". Fig. 2a and 2b have been updated to include method performance on these datasets.

a.

b.

Updated Fig. 2 (a) Summary heatmap of methods predicting spatial gene expression from H&E images highlighting key characteristics, and ranking their performances under each evaluation category. (b) Detailed heatmap of rankings of each method under each evaluation metric grouped by category.

In addition, we performed an additional experiment to explore the method’s transferability from low to high resolution data, using the HER2+ ST trained model (best fold) to directly predict Visium-HER2+ data. We evaluated and compared using the same metrics described in our original manuscript. The results are provided on page 6:

“Most methods showed limited effectiveness, except for HisToGene. HisToGene, originally trained using super-resolution with smaller tissue patches, allows the model to capture features in the scale that is more similar to Visium data. These results highlight the need for improved adaptation strategies to accommodate different platforms with varying imaging resolutions.”

2. While the manuscript assesses the accuracy of methods by measuring gene expression correlations within tissues, this approach may not be comprehensive enough. Considering the datasets include gold-standard spatial delineations in pathological images, I suggest incorporating additional metrics to enhance the thoroughness of the evaluations.

Response:

As suggested, we utilised the gold-standard delineations of the spatial regions in eight H&E images from the HER2+ ST dataset. We calculated the Adjusted Rand Index (ARI) for the clustering results based on all predicted SGE and tissue delineations. The ARIs for each method across the eight images have been added in Supplementary Fig. 3. The details have been incorporated into the

revised manuscript in the Results section on pages 4-5 under “Most methods can capture biologically relevant gene patterns from tissue images using ST datasets”:

“We used the predicted SGEs to perform K-means clustering and identify spatial regions in eight H&E images from HER2+ samples (Supplementary Fig. 3). Interestingly, the clustering results based on Ground Truth SGE did not always outperform those generated using predicted SGE. This suggests that the predicted SGE from H&E images captures additional imaging features from each spot and its surrounding tissue, providing a more comprehensive view of spatial patterns that enhances tissue region identification.”

Supplementary Fig. 3: K-means clustering of spatial regions based on eight samples from HER2+ ST dataset was performed using predicted gene expression of each method. The ground truth annotations are based on manual delineation by pathologists, while the ground truth SGE is derived from sequencing data. ARI was calculated between the ground truth annotations and the clustering results of each method. In sample B1, Hist2ST had the best performance, achieving the highest Adjusted Rand Index (ARI) of 0.36, followed by ST-Net with an ARI of 0.28 and THtoGene with 0.27. These methods outperformed the ground truth SGE, which had an ARI of 0.19.

3. The manuscript should discuss the limitations of the current methods in detail and demonstrate how integrating recommended new approaches with existing tools can address these issues.

Response:

We have expanded on the limitations of each method and added an additional column in Supplementary Table 2 named “Improvement Directions” to provide recommendations for new approaches to improve from the existing methods in the revised manuscript. We have also updated the summary of limitations and recommendations of the common issues in the Discussion section on page 10 as follow:

“Additionally, some methods struggle to handle densely expressed or small gene sets as they experience convergence difficulties or overfitting during training. Therefore, it is crucial to carefully select gene sets for training and prediction, combining biologically meaningful genes while balancing their sparsity to achieve optimal results. Moreover, improving the transferability or flexibility between different resolutions and cancer subtypes is essential to align with real-world applications. Using foundation models trained on a diverse range of tissues could help bridge this gap.”

4. The study evaluates method performance solely on tumor tissue sections. Given the significant heterogeneity of tumor tissues, this might not adequately reflect the methods' effectiveness. I recommend conducting evaluations across a broader range of tissue types.

Response:

We included additional tissue type by including non-tumour tissue from the Visium-Kidney dataset, which is believed to exhibit lower heterogeneity compared to tumour tissues (e.g. Visium-HER2+). Here, we evaluate the performance for non-tumor tissue by evenly dividing each tissue into four non-overlapping sections, and 4-fold cross validation was performed following the same settings as for the ST datasets. We have updated the results in Fig. 3f and added Supplementary Fig. 6. We have added the following in the manuscript:

Under Results section on page 6:

“When evaluating based on the Visium-Kidney dataset (Supplementary Fig. 6), a non-tumour tissue, Fig. 3f shows that DeepPT achieved the highest average correlation of HVG of 0.45 and SVG of 0.47, followed by ST-Net (HVG=0.31, SVG=0.36) and EGNv1 (HVG=0.25, SVG= 0.24). The strong performance of DeepPT and ST-Net (CNN-based methods) on non-tumour tissue is likely due to the low tissue heterogeneity of the sample, where patches containing a spot provided sufficient information for learning, making additional neighbourhood information less valuable due to the uniformity of surrounding tissues.”

Under Discussion section on page 9:

“This is particularly true for tissues with low heterogeneity, where global information tends to be less useful in images with smaller variations between patches.”

Updated Fig. 3f PCC violin and boxplots for each method in Visium-Kidney for all genes as well as for HVGs, SVGs and HSGs only. Significance between HVGs and all genes, SVGs and all genes, HSGs and all genes are calculated using Wilcoxon rank-sum test (The significance levels were defined as $^{\circ}p < 0.1$, $^*p < 0.05$, $^{**}p < 0.01$, $^{***}p < 0.001$).

Supplementary Fig. 6: Violin and boxplots of evaluation metrics for gene expression for each method in the Visium-Kidney dataset.

5. It would be beneficial to assess the precision of methods when handling data with varying degrees of gene expression matrix sparsity, spatial resolution, and gene counts to thoroughly evaluate their performance.

Response:

We thank the reviewer for the comment. We have designed a series of new experiments including our response to comment 1 for exploring and assessing the precision at varying levels of resolutions and gene matrix sparsity, while carefully considering the computational cost of applying this across all methods.

We explored various levels of gene expression matrix sparsity while training and testing models using Visium data. We concluded that while HSGs improve the model training process, they do not lead to better prediction results. Visualisations have been added in Supplementary Figs. 7 and 8, and the datasets' sparsity distribution is shown in Supplementary Fig. 9. The results are presented in Supplementary Figs. 10 and 11. Additionally, we have updated the Results section under "Spatial resolutions and gene matrix sparsity influence methods' performance" on page 6:

"In general, we found that a number of methods struggled with Visium data. These methods showed almost no variation in predicted gene expression across spots when training using Visium-Kidney and Visium-Hercep-Test2+, (Supplementary Fig. 7, 8) data, which did not happen with ST datasets. One potential reason is due to the difference in gene matrix sparsity, where Visium data typically has more densely expressed genes compared to ST datasets (Supplementary Fig. 9). We are able to resolve this issue for two methods (His2ST; THltoGene) by selecting higher sparsity genes for training (Supplementary Fig. 10, 11), but found that it did not improve overall prediction performance."

Additionally, in terms of gene counts, we evaluated methods' performance on smaller subsets of genes in our initial submission, including HVGs and SVGs, and concluded that methods performed better with fewer biologically meaningful genes. Further exploration of training effects with varying gene counts would require substantial computational resources, making it challenging to conduct within the scope of our current study.

Supplementary Fig. 7: Coefficient of variation for ground truth and predicted 992 HVGs, and ground truth and predicted 145 HSGs across spots in the Visium-Kidney dataset.

Supplementary Fig. 8: Coefficient of variation for ground truth and predicted 990 HVGs, and ground truth and predicted 274 HSGs across spots in the Visium-HER2+ dataset.

Supplementary Fig. 9: Gene expression matrix sparsity across different datasets. *ST-Visium-HER2+* refers to the sparsity of the gene set used for validating HER2+ ST trained models. *Visium-Hercep-Test2+* and *Visium-HER2+* represent the sparsity of gene sets used for training and prediction in the Visium Breast Cancer models, respectively. *Visium-Kidney* represents the sparsity of the gene sets in the Visium-Kidney dataset.

Supplementary Fig. 10: Violin and boxplots of the average PCC, MI, SSIM and AUC between the ground truth gene expression and predicted gene expression. Metrics measured from the test fold of a 4-fold CV, averaged over each gene across the Visium-Kidney dataset.

Supplementary Fig. 11: Violin and boxplots of the average PCC, MI, SSIM and AUC between the ground truth gene expression and predicted gene expression. Metrics measured from the test fold, averaged over each gene across the Visium-HER2+ dataset.

6. In the section "Characteristics of methods play a large role in gene-specific performance" the authors have not discussed which neural network architectures are best suited for addressing this issue. This section more closely resembles an ablation study and lacks a detailed analysis. Additionally, from Figure 3c, I am unable to obtain sufficient information to assess the similarity in performance clustering among methods.

Response:

We have provided a detailed analysis of which neural network architectures are best suited for this issue and clarified the findings from the clustering results in Fig. 3c (updated as Fig. 3e). We have edited the section "Characteristics of methods play a large role in gene-specific performance" in the Results section on page 5 to reflect these updates.

"Based on the predicted SGE performance in the ST datasets, we found that the methods which were members of the cluster comprising EGNv2, ST-Net, and DeepPT outperformed the others. This suggests that methods focusing on extracting image features within patches using CNN-based architectures are particularly well-suited for this task. The consistent distribution of prediction performance across genes highlights the influence of the structural design of these methods. This finding provides guidance for architecture selection in future method development."

Minor Comments:

1. Please verify the necessity of redeclaring '(Figure 1b)' at line 114 to avoid potential redundancy.

Response:

Thank you and we have removed it as suggested.

2. The statement at line 150, 'GNAS ($r = 0.47$) and FASN ($r = 0.47$),' appears inconsistent, as the correlation coefficients for these two genes by EGNv2 seem not to be the same according to Figure 2e. Please verify this data.

Response:

Thank you and we've modified FASN to the correct value. It now reads:

"Here, EGNv2 observed the highest correlations in genes GNAS ($r = 0.47$) and FASN ($r = 0.46$) in HER2+ ST dataset"

3. At line 155, 'MYL12B, with an average correlation of 0.25,' it is essential to clarify whether 'average' refers to the mean correlation across multiple samples or tests, and how it was calculated.

Response:

Thank you, we have clarified the term "average correlation" in the Materials and Methods section under "Gene expression prediction" in the revised manuscript. It now reads:

"Metrics under this category of evaluation were first calculated at an image level (e.g., correlation was measured for each gene per image), and then averaged over each patient (referred to as the average correlation of specific genes), then averaged over each gene."

4. Review the use of abbreviations for professional terms throughout the manuscript. Ensure terms like 'GNN' are introduced with both the full name and abbreviation on first mention and used consistently thereafter, avoiding repeated annotation of both the full name and abbreviation at lines 66 and 492.

Response:

Thank you, checked and removed as suggested.

Reviewer #2 (Remarks to the Author):

In this paper, the authors benchmark multiple methods for predicting spatial gene expression from histology data. Diverse metrics are used for evaluation, including not only prediction accuracy but also usability and computational efficiency. The paper provides valuable insights into the performance of the prediction models. However, there are areas that need major improvements.

1. For experiments on the HER2+ and CSCC datasets, it is unclear whether any subject appears in both the training and the testing sets, since each subject can contain multiple adjacent tissue sections.

Response:

We thank the reviewer for this comment. We would like to clarify that none of the adjacent tissue sections from the same patient were included in both the training and test sets. The data split was performed at the patient level. We have revised the manuscript to make this statement clearer.

In the Materials and Methods section under "Gene expression prediction" on page 17, lines 749-752:

"The SGE prediction of the ST datasets was evaluated using 4-fold cross validation (CV). The HER2+ and cSCC ST datasets were split into four folds, ensuring that adjacent tissue sections from the same patient were grouped within the same fold. In each iteration of the

CV, two folds were used as a training set, one fold used as a validation set and the remaining fold used as a test set.”

For the TCGA dataset, are training and testing samples both from TCGA, or are training and testing performed on different studies?

Response:

The TCGA dataset was used as an external test set. The prediction models applied to the TCGA samples were selected from the best-performing models, based on the 4-fold CV trained on the HER2+ ST dataset. We added under the Results section under “Comparing the translational potential across methods using TCGA-BRCA data” on page 6, lines 250-252:

“Next, we applied the best fold models, selected based on validation set performance and pretrained on the HER2+ ST dataset, to predict bulk RNA-Seq gene expression patterns for these images.”

2. For predicting spatial gene expression from histology images, an important aspect is cross-subject and cross-study prediction, but this type of prediction seems to be not rigorously evaluated in this paper. For the TCGA dataset, evaluation is only performed at the bulk-tissue level, which does not measure the prediction accuracy of spatial gene expression. For the HER2+ and CSCC datasets, it is not clear whether the authors have performed cross-study evaluation. Even if so, the experiments are still limited to two datasets. Including only 3 datasets is a major limitation for a benchmarking paper. Given the public availability of numerous spatial transcriptomic datasets, it would greatly improve this paper's contribution to spatial omic research if multiple independent datasets are included and evaluation is performed by using some datasets for training while others for testing.

Response:

Thank you for the comment. We did not use TCGA to directly evaluate the expression prediction as it does not provide ground truth SGE. Instead, TCGA was used for clinical translational evaluation as a cross-study/external dataset to assess the models trained on HER2+ ST.

While there is an increased number of spatial omics data available, matched H&E and omics availability remains limited. To complement the TCGA study, we have now added an experiment where we:

- train using the HER2+ ST datasets and select the best-fold model using validation data (same models applied to TCGA); and
- test using the Visium-HER2+ dataset

In general, most methods found predictions across different resolutions challenging and performance is generally weak as expected. We added Supplementary Fig. 13 and the results is added to the revised manuscript (please see response to Reviewer #1 Comment 1).

Additionally, we have also extended to another cross-study prediction using the similar resolution datasets:

- train using the Visium-Hercep-Test2+ dataset; and
- test using the Visium-HER2+ dataset

Most methods found predictions across different cancer subtypes challenging. We updated the results in Fig. 3g and added Supplementary Fig. 12. The following results is added to the revised manuscript on page 6:

“EGNv1 demonstrated significant improvement in correlation using HVGs and SVGs of 0.11 and 0.13, respectively (Fig. 3g), their performance is in general weaker than the within study evaluation. This suggests that models trained on a specific breast cancer subtype struggle to generalise to others, as the biological characteristics differ between subtypes.”

Supplementary Fig. 13: Violin and boxplots of gene-level correlations between ground truth and predicted gene expression across two adjacent tissue slides from Visium-HER2+. The models were trained on the HER2+ ST dataset, with the best-performing model selected based on 4-fold cross-validation.

Fig. 3g: PCC violin and boxplots for each method in Visium-HER2+ dataset for all genes as well as for HVGs, SVGs and HSGs only. Significance between HVGs and all genes, SVGs and all genes, HSGs and all genes are calculated using Wilcoxon rank-sum test (The significance levels were defined as $^{\circ}p < 0.1$, $^*p < 0.05$, $^{**}p < 0.01$, $^{***}p < 0.001$).

Supplementary Fig. 12: Violin and boxplots of evaluation metrics for gene expression for each method in the Visium-HER2+ dataset.

3. Fig 2a-b: It is useful to summarize the evaluations of all methods across various metrics in one chart, but it is unclear whether each entry of the table is a subjective score or computational evaluation. Overall, the numerical rigor of Fig2a-b needs to be improved.

Response:

We have clarified the entries of Fig. 2a-b in Materials and Methods section under “Performance evaluation” on page 17, lines 723-725:

“The blank spaces in the rows indicate that the methods were not applicable to those categories. Each entry in Fig. 2a and 2b represents the ranking of the corresponding metric across all methods.”

4. Fig 2a-b: Which datasets are used for evaluating the computational efficiency of each method?

Response:

We selected one image from HER2+ ST dataset (Sample A1) for computational efficiency evaluation. We have revised in the Materials and Methods section under “Performance” on page 20, lines 860-862 and it now reads:

“For calculating performance, memory and time taken to train one image from HER2+ dataset (Sample A1) over 10 epochs were used as a unit for comparison across methods (Supplementary Fig. 19).”

5. For each panel in Figure 2, which dataset is used for evaluation? It is also not clearly indicated whether training and testing are performed in the same subject/study or across different subjects/studies.

Response:

This has been clarified in the Materials and Methods section under “Performance evaluation” on page 17:

“Metrics of predicted SGE from the test samples of HER2+ and cSCC ST datasets were used to build the “ST Gene Expression Prediction” category, while the “Visium Gene Expression Prediction” category was based on the prediction results from Visium-HER2+ and Visium-Kidney test samples. Methods that failed to perform were ranked lowest in the corresponding metrics. For the “Model Generalisability” and “Clinical Translational Impact” categories, metrics were derived from analyses using TCGA data.”

6. Fig 2b: This is a very useful chart, and visualizing ranks is also a good approach. However, the legends are not very visually distinguishable, since ranks are represented by the shape and size of the markers. It would be easier for readers to find the best method for each column if the ranks are color-coded (e.g. by using a heat map).

Response:

Thank you for the suggestion. We have updated the colours for each category, using darker colours to indicate higher ranks and better performance.

7. Fig 2a, third column: What does "reimplemented" mean here?

Response:

Apologise for the typo. “Reimplemented” means the code was not publicly available, so we implemented the methods by referring to their papers. We have now clarified in the Material and Methods section on page 17, lines 732-733:

“Reimplementation” refers to situations where a method had to be implemented based on the original paper because the code was not publicly available. “

8. Fig 2d: What does each point in the box plot represent? A gene?

Response:

Each point represents a gene. We have now clarified this in Materials and Methods under “Gene expression prediction” on page 18, lines 763-766 as follows:

“Metrics under this category of evaluation were first calculated at an image level (e.g., correlation was measured for each gene per image), and then averaged over each patient (this is referred to as average correlation of specific genes), then averaged over each gene.”

9. Fig 3e: What does the heat map look like if the genes are sorted by mean or variance?

Response:

Fig. R1 is sorted by the mean value of the genes, which doesn’t change the clustering results; the same result applies to sorting by gene variance.

Fig. R1: Heatmap of average correlation of each gene in HER2+ ST dataset and each method sorted by log of the mean of each gene. The log of the mean and variance of each gene are coloured above the heatmap.

10. If prediction is performed for each spot, how SSIM is computed for the predicted and ground truth spatial gene expression, since standard SSIM is only applicable to pixelized data?

Response:

While SSIM is traditionally used for pixel-based data, we adapted the concept for spot-based spatial gene expression (SGE) by treating each spot as a "pixel" in the spatial grid. In our case, the spatial gene expression matrix is structured such that each spot corresponds to a location in the tissue, similar to how each pixel corresponds to a location in an image. We have updated Table 2 in the revised manuscript to clarify the definition.

11. I might have missed this but how are SVG and HVG defined

Response:

We defined SVG and HVG in our initial submission in Materials and Methods under "Gene expression prediction" on page 18, lines 773 - 782.

Reviewer #3 (Remarks to the Author):

The authors benchmarked several methods for predicting gene expression data from histological images. Overall, the insights and conclusions are too general to merit publication in Nature Communications. Additionally, the authors used only a few tumor types or samples for benchmarking and did not consider the recent surge in new papers in this field, such as iStar 1.

1 Zhang, D. et al. Inferring super-resolution tissue architecture by integrating spatial transcriptomics with histology. Nature Biotechnology, doi:10.1038/s41587-023-02019-9 (2024).

Response:

We would like to reiterate that most evaluation criteria to date have solely focused on assessing predictive performance. This is the first benchmarking paper that systematically applied several methods on H&E data alone, investigated data characteristics that affect prediction performance, and assessed the impact of these predictions performance on downstream applications (e.g. survival outcome). Again, the key findings provide several important observations that are crucial to the advancement of this emerging field. This includes:

- acknowledging that average correlation alone proved to be an inadequate metric for assessment in isolation;
- level of expression, spatial variability and variability across tissue types and patients all impact the capacity to predict spatially resolved expression from histology;
- the complexity of deep learning architecture did not confer superiority in either spatial gene expression prediction or translational potential categories.

These observations are further strengthened with additional comparison and new experiments described in response to Reviewer #1 and #2.

Regarding the number of papers, we have conducted an extensive literature search prior to our initial submission on 6th January 2024. The list of papers is documented in the Materials and Methods section under “Benchmarked methods selection criteria”, with 11 methods included in the final benchmarking. The inclusion and exclusion criteria can be found on pages 10-11 of the revised manuscript.

We have made additional exploration relating to different tissues types, spatial resolutions and datasets in the revised manuscripts. We have now added iStar into our evaluation program and updated the method descriptions and results of iStar to our revised manuscript in all relevant sections. We have implemented iStar and performed the following new experiments:

- Training and testing on HER2+ ST (4-fold CV)
- Training and testing on cSCC ST (4-fold CV)
- Training and testing on Visium-Kidney (4-fold CV)

In summary, iStar did not outperform the original list of methods we benchmarked. It achieved average performance on Visium data, which was better compared to its performance on ST data.

Practical implementation of iStar requires consistency in image size, as well as image and spot resolutions, when training and testing across datasets. After rescaling to make a 16×16 patch represent a single cell, images from datasets with different resolutions vary significantly in size. Since image embeddings need to have the same dimension to be trained together, resizing can lead to information loss, requiring input images to have consistent dimensions. This requirement posed challenges when attempting to include TCGA and Visium-HER2+ as external validation data or perform cross-study analyses between the two breast cancer data Visium-Hercep-Test2+ and Visium-HER2+. The details of the method have been added to Supplementary Method 1.

Reference

Zhang, Daiwei, et al. Inferring super-resolution tissue architecture by integrating spatial transcriptomics with histology. *Nature biotechnology* **42**, 1372–1377 (2024)

REVIEWER COMMENTS

Reviewer #1 (Remarks to the Author):

The authors revised all my concerns.

Response: We thank the reviewer for the comment.

Reviewer #2 (Remarks to the Author):

While the authors have addressed most my comments, my biggest concern remains that only a few SRT datasets are included in this study. Even with the newly added experiments, there are only 5 SRT datasets in total. By contrast, there were at least a dozen of Visium datasets (including H&E images) on 10x Genomics' website alone one year ago, and half a year ago two more publicly sourced multi-study SRT meta-datasets were curated (HEST-1k and STimage-1K4M), each containing about 1000 SRT+H&E slides. Extending the authors' current study setup for Visium to more datasets, which have the same formats, should be straightforward. Given that generalizability is a key bottleneck in current algorithms for predicting SRT from histology, more datasets need to be included to make the results more generalizable to meet the standard of Nature Communications.

Response: We thank the reviewer for the valuable suggestion regarding the new STimage-1K4M and HEST-1k datasets and agree that these resources provide exciting opportunities for benchmarking spatial transcriptomic technologies. Although we recognise their potential to enhance our current benchmarking, both were released in June and December 2024, respectively, and therefore were not fully accessible during our submission timeline (January–October 2024). While adding more data might broaden our analysis, our assessment indicates that it would not significantly alter or contradict our current conclusions, but would significantly delay the sharing of our valuable conclusions with the community. Specifically, cross-patient evaluations on multiple ST and Visium datasets demonstrated the potential of spatial gene expression (SGE) prediction methods for disease related patient insights but highlighted significant limitations regarding the interpretation and reliability of predictions. Meanwhile, our in-sample evaluations on Visium healthy kidney tissue revealed moderate predictive performance for tissues with low heterogeneity. These results collectively capture a range of outcomes that the new datasets provide.